# SAFE BAYESIAN OPTIMIZATION FOR COMPLEX CONTROL SYSTEMS VIA ADDITIVE GAUSSIAN PROCESSES

## ABSTRACT

Controller tuning and optimization have been among the most fundamental problems in robotics and mechatronic systems. The traditional methodology is usually model-based, but its performance heavily relies on an accurate mathematical system model. In control applications with complex dynamics, obtaining a precise model is often challenging, leading us towards a data-driven approach. While various researchers have explored the optimization of a single controller, it remains a challenge to obtain the optimal controller parameters safely and efficiently when multiple controllers are involved. In this paper, we propose SAFECTRLBO to optimize multiple controllers simultaneously and safely. We simplify the exploration process in safe Bayesian optimization, reducing computational effort without sacrificing expansion capability. Additionally, we use additive kernels to enhance the efficiency of Gaussian process updates for unknown functions. Hardware experimental results on a permanent magnet synchronous motor (PMSM) demonstrate that compared to existing safe Bayesian optimization algorithms, SAFECTRLBO can obtain optimal parameters more efficiently while ensuring safety.

## 1 INTRODUCTION

Optimizing the parameters of complex systems with multiple controllers is a challenging task, particularly in configurations like the cascade feedback control architecture commonly used in motor control, as well as advanced controllers involving feedforward control, disturbance observers (DOB) (Jung & Oh, 2022), and active disturbance rejection control (ADRC) (Cao et al., 2024). For example, in permanent magnet synchronous motor (PMSM) control, field-oriented control (FOC) is widely employed (Gabriel et al., 1980; Lara et al., 2016; Wang et al., 2016). The closed-loop FOC configuration involves three independent proportional-integral (PI) controllers, each requiring the tuning of two control gains. These six gains have different parameter ranges and must be optimized simultaneously to achieve the best control performance. Each adjustment of the parameter combination requires an evaluation process lasting several minutes and also demands significant expertise from a control engineer. Therefore, there is a strong need for an efficient, automatic optimization approach.

Traditional automatic tuning and optimization methods rely on simplified reduced-order models with assumptions such as linearity. These assumptions, along with modeling errors, often lead to suboptimal performance of controllers in real-world systems (Berkenkamp et al., 2016). Meanwhile, motion data from real-world systems operating under suboptimal conditions often contain valuable information that traditional model-based methods fail to fully exploit. Data-driven control optimization addresses this limitation by directly leveraging the information in the motion data to optimize controller parameters. It typically models the system's performance as a function of controller parameters and then explores the optimal parameter iteratively. In this line of research, various algorithms have been designed, with gradient-based algorithms being among the most popular approaches; however, they require accurate gradient estimates (Li et al., 2024), which can be challenging to obtain with noisy experimental measurements and often lead to convergence at local optima. Genetic algorithms, on the other hand, require extensive testing, making them impractical for real-world applications (Davidor, Jan. 1991).

Bayesian optimization (BO) (Mockus, 2012) offers a solution to these challenges by modeling the system's performance function using a Gaussian process (GP) (Rasmussen & Williams, 2006). In this framework, each controller parameter combination is associated with a performance value repre-

sented by a Gaussian distribution, which includes noise measurements. However, the BO procedure iteratively tests parameters with the highest uncertainty, often evaluating potentially unsafe controller parameters, which may lead to system instability. Therefore, controller optimization requires the use of a *safety-aware* BO algorithm, and some relevant work is introduced as follows.

**Related work.** The SAFEOPT (Sui et al., 2015) and STAGEOPT (Sui et al., 2018) algorithms were among the first to address the safety concerns in Bayesian optimization (BO). They introduce the concept of a safe set to avoid evaluating parameters that do not meet a predefined safety threshold, thus ensuring safety. Subsequent work includes (Turchetta et al., 2019b; Bottero et al., 2022; Fiedler, 2023; Fiedler et al., 2024; Whitehouse et al., 2023), focusing on algorithm performance improvement or theoretical analysis. Berkenkamp et al. (2016) applied SAFEOPT to quadrotor controller tuning, validating its practical effectiveness. They employed the Matérn kernel with $\nu = 3/2$ as the covariance function of the Gaussian process, which is effective mainly for low-dimensional problems (Bengio et al., 2005). In Berkenkamp et al. (2016), the $x$, $y$, and $z$-axis PD controllers of the quadrotor were optimized separately, with each controller having only two parameters. Similarly, Fiducioso et al. (2019) automated the tuning of only two parameters for a room temperature controller in a simulator. Moreover, SAFEOPT employs a maximum uncertainty sampling acquisition function to balance exploration and exploitation, but this can cause fluctuations in the evaluated objective function values, making it difficult to converge. While the stage-wise algorithm in Sui et al. (2018) guarantees convergence in the optimization phase, it still does not improve efficiency in higher dimensions.

Djolonga et al. (2013) suggested that high-dimensional problems could be decomposed into several lower-dimensional subspaces for optimization. Building on this, Kirschner et al. (2019) proposed the LINEBO algorithm, which decomposes a high-dimensional space into multiple one-dimensional subspaces for safe BO within each subspace. However, this method often requires hundreds or even over a thousand iterations to find an optimal solution. While feasible for general optimization problems where performance evaluation can be easily simulated, this approach is less practical for real-world experiments, such as those in complex control optimization.

There are two main differences between the complex control optimization tasks we focused on and the general high-dimensional optimization problems tackled by LINEBO (Kirschner et al., 2019), making it less effective for our control problems. First, complex control optimization typically involves problems with dimensions ranging from low to moderate (6 to 20 parameters). For instance, the electric motor field-oriented control (FOC) system has three PI controllers with six parameters (Gabriel et al., 1980; Lara et al., 2016; Wang et al., 2016); the quadrotor system has six control parameters for three axes, or twelve if angle control is included (Berkenkamp et al., 2016; Yuan et al., 2022); and the gantry system consists of three axes with outer-loop P controllers and inner-loop PI controllers, resulting in six to nine parameters (Rothfuss et al., 2023; Wang et al., 2022; 2023). In contrast, the problems studied by Kirschner et al. (2019) typically involve 10 to 100 parameters, making the problem scale significantly different. Second, in control optimization, each iteration involves applying the controller parameters to the hardware to obtain performance and safety evaluations, which can take several minutes or longer. Additionally, repeated iterations cause wear on the hardware, making an excessive number of iterations unacceptable. In contrast, the problems in Kirschner et al. (2019) allow for hundreds or even thousands of iterations. Therefore, a more efficient safe optimization algorithm is required for complex control problems.

As noted by Bengio et al. (2005), the locality of Gaussian kernels limits Gaussian process models in capturing non-local structures. To address this, Duvenaud et al. (2011) introduced additive Gaussian processes, creating a high-dimensional additive structure for Gaussian kernels, significantly improving the Gaussian process's capability to model high-dimensional unknown functions. Studies by Rolland et al. (2018); Kandasamy et al. (2015); Mutny & Krause (2018); Bardou et al. (2024) show that additive Gaussian processes increase efficiency in high-dimensional BO. However, experimental validation of these methods remains limited, and their integration with safety constraints has not been fully explored.

**Our contributions.** Given the traits of multi-parameter complex control systems, our main contributions are as follows: 1) We introduced additive kernels to *safety-aware* BO for the first time and theoretically evaluated its convergence under safety constraints. These kernels accommodate different signal variances and lengthscales across dimensions, making them well-suited for complex

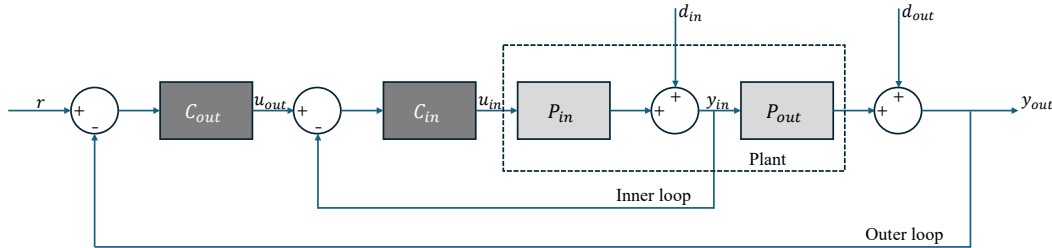

Figure 1: A block diagram for a 2-layer cascade system. The dark grey blocks represent controllers, and the light grey blocks represent plants.

control systems where controllers have varying parameter ranges. 2) We simplified the computation of the potential expander set used in previous safe BO methods, reducing the computational burden while preserving safe exploration capabilities, as proved theoretically and demonstrated experimentally. 3) We conducted comprehensive simulations on synthetic functions and control system hardware experiments. The simulations showed that our proposed method is competitive with other safe BO methods in terms of optimization performance, while the hardware experiments validated its practical applicability against several safe BO algorithms in a real-world random error environment.

## 2 PROBLEM STATEMENT

Similar to Khosravi et al. (2023), the safe optimization problem for complex cascade systems is considered. Cascade systems have multiple controllers, and the output of the outer loop controller serves as the input of the inner loop controller. Consider the discrete-time proportional-integral (PI) control law:

$$u_k = k_p \cdot (r_k - y_k) + k_i \cdot \sum_{t=0}^{k} (r_k - y_k), \tag{1}$$

where $u_k$ is the control action in time step $k$, $y_k$ is the plant output, $r_k$ is the reference signal, and $(k_p, k_i)$ are the control gains. In a 2-layer cascade system (Figure 1), the control laws for both layers will be:

$$u_k^{in} = k_p^{in} \cdot (u_k^{out} - y_k^{in}) + k_i^{in} \cdot \sum_{t=0}^{k} (u_k^{out} - y_k^{in}), \quad u_k^{out} = k_p^{out} \cdot (r_k - y_k^{out}) + k_i^{out} \cdot \sum_{t=0}^{k} (r_k - y_k^{out}) \tag{2}$$

In a general form, denote the outermost layer as layer $0$, and the $n$th inner layer as layer $n$, then the control action $u_k$ in layer $n$ is a function of the plant output $y_k$ in all layers from layer $0$ to layer $n$, the reference signal $r_k$, and the controller parameters $a$:

$$u_k^n = g((y_k^0, y_k^1, ..., y_k^n), r_k, a), \tag{3}$$

where $a \in \mathcal{A}$, and $\mathcal{A}$ is the domain for possible controller parameters. The controller's performance measure depends on how well it accomplishes its objective. Instead of modeling complex systems, performance measurement is modeled as a function of controller parameters, $J(a) : \mathcal{A} \longmapsto \mathbb{R}$, and all constraints are modeled as functions of controller parameters, $G(a) : \mathcal{A} \longmapsto \mathbb{R}$. Both $J(a)$ and $G(a)$ are evaluated on the systems, using cost functions such as Integral Square Error (ISE), Integral Absolute Error (IAE), or Integral Time-weighted Absolute Error (ITAE).

We aim to solve a sequential decision problem where we seek to find $a$ that maximizes $J(a)$ while ensuring that all $G(a)$ satisfy the safety constraints. Safety considerations are included in $G(a)$. Assuming the availability of an initial safe controller and its performance, $(a_0, J(a_0))$, a sequence of parameters $a_1, a_2, ..., a_n \in \mathcal{A}$ is selected, and the noisy performance measurement $J(a_n) + d_n$ is obtained after each selection. During the evaluation, $G(a_n) \geq h$ must hold with high probability for all $G(a_n)$, where $h$ is the safety threshold. In control applications, the goal is typically to find the optimal controller parameters that result in a faster transient response, minimal overshoot, and reduced steady-state error, while ensuring that physical quantities such as current, voltage, and power remain within safe limits during evaluation and that the system remains stable at all times.

## 3 SAFE BAYESIAN OPTIMIZATION

In previous studies, safe Bayesian optimization (BO) methods were designed to address general safe sequential decision-making problems. These methods use the Gaussian processes (GP) to approximate unknown performance and safety functions. By defining an appropriate covariance function $k(\mathbf{a}_i, \mathbf{a}_j)$, GPs can combine past observations to predict the mean and variance of the value of the objective function at unobserved points:

$$\mu_n(\mathbf{a}) = \mathbf{k}_n(\mathbf{a})(\mathbf{K}_n + \mathbf{I}_n \sigma_\omega^2)^{-1} \tilde{\mathbf{J}}_n, \quad \sigma_n^2(\mathbf{a}) = k(\mathbf{a}, \mathbf{a}) - \mathbf{k}_n(\mathbf{a})(\mathbf{K}_n + \mathbf{I}_n \sigma_\omega^2)^{-1} \mathbf{k}_n^T(\mathbf{a}), \quad (4)$$

where $\tilde{\mathbf{J}}_n = [\tilde{J}(\mathbf{a}_1), ..., \tilde{J}(\mathbf{a}_n)]^T$ is the vector of noisy performance measurements, the matrix $\mathbf{K}_n$ has entries $[\mathbf{K}_n]_{(i,j)} = k(\mathbf{a}_i, \mathbf{a}_j)$, and the vector $\mathbf{k}_n(\mathbf{a}) = [k(\mathbf{a}, \mathbf{a}_1), ..., k(\mathbf{a}, \mathbf{a}_n)]$. $k(\mathbf{a}_i, \mathbf{a}_j)$ is also called the kernel of the GPs.

The effectiveness of safe BO relies on two key assumptions. First, the objective functions have bounded norms in their Reproducing Kernel Hilbert Spaces (RKHS) associated with the GPs, and second, the objective functions are Lipschitz-continuous. The RKHS norm in GP is directly related to the smoothness and variability of the functions. Bounded norms allow for tighter confidence intervals around the GP predictions. Lipschitz continuity provides a mathematical basis for estimating how much the safety function may change with a given change in input. When both assumptions hold, after computing the mean and variance of each point in the objective function, the value of the objective function is constrained within the GP's confidence interval. Specifically, the upper and lower bounds of the confidence interval are:

$$u_n(\mathbf{a}) = \mu_{n-1}(\mathbf{a}) + \beta_n \sigma_{n-1}(\mathbf{a}), \quad l_n(\mathbf{a}) = \mu_{n-1}(\mathbf{a}) - \beta_n \sigma_{n-1}(\mathbf{a}), \quad (5)$$

where $\beta_n$ is a variable defining the confidence interval. Then, acquisition functions, such as the GP-UCB method (Srinivas et al., 2010) that maximizes the upper confidence interval, or the method of maximizing the Gaussian process variance used in SAFEOPT (Berkenkamp et al., 2016), select the next predicted parameter $\mathbf{a}$ based on the confidence interval.

The most important feature of previous safe BO algorithms (Berkenkamp et al., 2016; Sui et al., 2018; Turchetta et al., 2019a) is their definition of the safe set, $\mathcal{S}_n = \{a \in \mathcal{A} \mid \forall i, \ l_n^{(i)}(a) \geq h_i\}$, which contains all the parameters that have high probabilities of getting the values of safety functions $G(a)$ above the safe thresholds $h_i$. Another important concept is the potential expander set, $\mathcal{E}_n = \{a \in \mathcal{S}_n \mid \forall i, \ \exists a' \in \mathcal{A} \setminus \mathcal{S}_n \text{ such that } l_{n,(a,u_n(a))}(a') \geq h_i\}$, which contains parameters that could expand $\mathcal{S}_n$ after a new iteration. Some algorithms also introduce the potential maximizer set, $\mathcal{M}_n = \{a \in \mathcal{S}_n \mid u_n^{(1)}(a) \geq max_{a' \in \mathcal{S}_n} l_n^{(1)}(a')\}$, which includes parameters that could obtain the optimal performance measurement $J(a)$. By restricting the parameters $\mathbf{a}$ selected for evaluation in each iteration to $\mathcal{S}_n$, previous safe BO algorithms ensure a high probability of not violating safety constraints during the optimization process.

## 4 SAFECTRLBO

### 4.1 ADDITIVE GAUSSIAN KERNELS

Despite advancements in high-dimensional safe BO methods, such as the SWARMSAFEOPT algorithm in Berkenkamp et al. (2016) and the LINEBO algorithm (Kirschner et al., 2019), the squared-exponential kernels used in these work have limited information acquisition efficiency in the parameter space. According to Srinivas et al. (2010), for a $d$-dimensional problem, the maximum information gain $\gamma_T$ for squared-exponential kernels conforms $\gamma_T = \mathcal{O}((logT)^{d+1})$, while for the $d$-dimensional Bayesian linear regression case conforms $\gamma_T = \mathcal{O}(dlogT)$. Therefore, we built upon the idea from additive Gaussian processes (Duvenaud et al., 2011), implementing high-dimensional additive structures to the original Gaussian kernels to achieve a lower $\gamma_T$.

The high-dimensional additive kernels for each order are formed by summing combinations of base kernels, where the base kernels are one-dimensional Gaussian kernels, $k(\mathbf{a}_i, \mathbf{a}_j) = \exp\left(-\frac{\|\mathbf{a}_i - \mathbf{a}_j\|^2}{2\sigma^2}\right)$. Let $z_i$ represent the base kernel for the $i^{\text{th}}$ dimension, then additive kernels

for different orders in a $D$-dimensional parameter space can be designed:

$$k_{add_1}(\mathbf{a}, \mathbf{a}') = \sum_{i=1}^{D} z_i, \quad k_{add_2}(\mathbf{a}, \mathbf{a}') = \sum_{i=1}^{D-1} \sum_{j=i+1}^{D} z_i z_j, \quad k_{add_n}(\mathbf{a}, \mathbf{a}') = \sum_{1 \le i_1 < i_2 < \ldots < i_n \le D} \prod_{d=1}^{N} z_{i_d} \tag{6}$$

The RKHS boundedness and Lipschitz continuity of the additive kernels are given in Appendix D.1. These properties ensure the additive kernels satisfy the two assumptions in safe BO.

## 4.2 ACQUISITION FUNCTIONS

In previous safe BO methods, the potential expander set $\mathcal{E}_n$ is often complicated. Consequently, in various implementations (Sui et al., 2015; Berkenkamp et al., 2016), candidates with smaller variances are typically filtered out, and only the outermost candidates with large uncertainty are considered. We provide the following definitions to formalize this explanation:

- Set of safe boundary points $\mathcal{B}_n$: $\mathcal{B}_n = \{a \in \mathcal{S}_n \mid l_n^{(i)}(a) = h_i\}$, is the subset of $\mathcal{S}_n$ where at least one constraint's lower confidence bound equals its safety threshold;

- Outermost evaluated safe points $a_{\mathrm{oes}}$: $\exists a_{\mathrm{sb}} \in \mathcal{B}_n$, $a_{\mathrm{oes}} = \arg\min_{a \in \mathcal{S}_n^{\mathrm{eval}}} \|a - a_{\mathrm{sb}}\|$, where $\mathcal{S}_n^{\mathrm{eval}}$ is the set of evaluated safe points in $\mathcal{S}_n$. For one $a_{\mathrm{oes}}$, it is an evaluated safe point such that for at least one safe boundary point $a_{\mathrm{sb}}$, $a_{\mathrm{oes}}$ is the closest evaluated safe point to $a_{\mathrm{sb}}$ in Euclidean distance.

- Outermost region $O_n$: $O_n = \bigcup_{a_{\mathrm{sb}} \in \mathcal{B}_n} \{\lambda a_{\mathrm{sb}} + (1 - \lambda) a_{\mathrm{oes}}(a_{\mathrm{sb}}) \mid \lambda \in [0, 1]\}$, is defined as the union of regions between each safe boundary point and its corresponding outermost evaluated safe point. In higher dimensions, this represents all points lying on the straight line segments connecting each $a_{\mathrm{sb}}$ to its nearest $a_{\mathrm{oes}}$.

$\mathcal{E}_n$ is simplified to $O_n$ in previous implementations, but when using additive kernels for optimization, calculating $O_n$ is still very time-consuming. In SAFECTRLBO, the calculation of $O_n$ is further simplified to $\mathcal{B}_n$, according to the following theorem:

**Theorem 4.1.** *When the outermost region $O_n$ is sufficiently large, within $O_n$, the point with maximum predictive uncertainty $\sigma_n^2(\boldsymbol{a})$ lies on the safe boundary $\mathcal{B}_n$.*

That is, obtaining the safe expander candidate with the largest uncertainty in $\mathcal{B}_n$ yields the same result as finding the candidate with the largest uncertainty in $O_n$, when the outermost region $O_n$ is sufficiently large. This leads to our acquisition function in the safe expansion stage, $a_n = \arg\max_{a \in \mathcal{B}_n} \sigma_n(a)$, where the selected parameters are consistent with those obtained by previous safe BO methods. We verify this through theoretical analysis and experiments in Appendix D.2 and E.

The complete procedure of SAFECTRLBO is shown in Algorithm 1. The additive kernels $k_{addD}$ are the sum of the additive kernels of various orders. We employ a stagewise iteration strategy: after expanding the safe set, we proceed to find the maximum value of the objective function. In the second stage, we choose GP-UCB as the acquisition function, and its convergence properties are discussed in Section 5.

## 5 THEORETICAL RESULTS

In this section, We present two theorems to analyze the theoretical effectiveness of SAFECTRLBO. Theorem 1 ensures the convergence of the safe expansion stage (line $2-8$ in Algorithm 1) in finite time, and Theorem 2 guarantees the convergence of the maximization stage (line $9-14$ in Algorithm 1) in finite time.

**Definition 5.1.** *In the safe expansion stage, an $\epsilon$-reachable safe region $\mathcal{R}_\epsilon$ is the safe set $\mathcal{S}_{t^*}$ obtained when $\exists \epsilon > 0, \forall i, max_{a \in \mathcal{B}_{t^*}} 2\beta_{t^*} \sigma_{t^*-1}^{(i)}(a) \le \epsilon$.*

**Definition 5.2.** *In the maximization stage, an $\zeta$-optimal function value $f(a_{opt})$ is obtained when the regret $r_t$ satisfies $\exists \zeta > 0, r_t = f(a^*) - f(a_{opt}) \le \zeta$.*

---

**Algorithm 1** SAFECTRLBO

---

**Inputs:** Controller parameter domain $\mathcal{A}$
Safe GP prior for performance function and safety functions $j$, $g_i$, $i \in \{2, \ldots, m\}$
Safe thresholds $h_i$, $i \in \{1, \ldots, m\}$
Additive kernels for performance and safety $k_{addD}$
Initial, safe controller parameters and its noisy performance measurement $(a_0, \tilde{J}(a_0))$
Stage switching time $T_0$

1: Initialize GP with $(a_0, \tilde{J}(a_0))$
2: **for** $n = 1, 2, \ldots, T_0$ **do**
3:     $\mathcal{S}_n \leftarrow \{a \in \mathcal{A} \mid l_n^i(a) \geq h_i\}$, $i \in \{1, \ldots, m\}$
4:     $\mathcal{B}_n \leftarrow \{a \in \mathcal{S}_n \mid l_n^i(a) = h_i\}$, $i \in \{1, \ldots, m\}$
5:     $a_n \leftarrow \arg\max_{a \in \mathcal{B}_n} \sigma_{n-1}^1(a)$
6:     Obtain noisy measurement $\tilde{J}(a_n)$
7:     Update GP with $(a_n, \tilde{J}(a_n))$
8: **end for**
9: **for** $n = T_0 + 1, \ldots$ **do**
10:     $\mathcal{S}_n \leftarrow \{a \in \mathcal{A} \mid l_n^i(a) \geq h_i\}$, $i \in \{1, \ldots, m\}$
11:     $a_n \leftarrow \arg\max_{a \in \mathcal{S}_n} u_n^1(a)$
12:     Obtain noisy measurement $\tilde{J}(a_n)$
13:     Update GP with $(a_n, \tilde{J}(a_n))$
14: **end for**

---

**Theorem 5.1.** *In the exploration stage of Algorithm 1, suppose the safety function $g_i$ satisfies $||g_i||_k^2 \leq B$ and is $L_i$-Lipschitz continuous, the noise at iteration $t$, $n_t$, is $R$-sub-Gaussian, the GP confidence level is $1 - \delta$. $\beta_t = R\sqrt{2(\gamma_{t-1} + 1 + log(1/\delta))} + B$, where $\gamma_t$ is the maximum information gain. Assume any $\epsilon > 0$, let $t^*$ be the smallest positive integer satisfying:*

$$t^* \geq C \left( \frac{\beta_{t^*} \sqrt{d}}{\epsilon} \right)^d.$$

*where $C$ is a constant depending on the problem parameters, $d$ is the dimension of the domain $\mathcal{A}$, then for all $t \geq t^*$, the safe set $S_t$ includes all points in the $\epsilon$-reachable safe region $\mathcal{R}_\epsilon$ with probability at least $1 - \delta$.*

We count $t$ from the beginning of the safe expansion stage and let $T_0 = t^*$ (line 2 in Algorithm 1). After $t^*$ iterations, the algorithm has fully explored the $\epsilon$-reachable safe region with high probability $(1 - \delta)$, and can proceed to the maximization stage. The complete proof of Theorem 5.1 is presented in Appendix D.3.1.

**Theorem 5.2.** *In the maximization stage of Algorithm 1, suppose the performance function $f$ satisfies $||f||_k^2 \leq B$ and is $L$-Lipschitz continuous, the noise at iteration $t$, $n_t$, is $R$-sub-Gaussian, the GP confidence level is $1 - \delta$. $\beta_t = R\sqrt{2(\gamma_{t-1} + 1 + log(1/\delta))} + B$, where $\gamma_t$ is the maximum information gain. Assume any $\zeta > 0$, let $T^*$ be the smallest positive finite integer that satisfies the inequality:*

$$\frac{C_\gamma d \ln T^*}{T^*} \left[ B + R\sqrt{2 \left( C_\gamma d \ln T^* + 1 + \ln \left( \frac{1}{\delta} \right) \right)} \right]^2 \leq \frac{\zeta^2}{4}.$$

*where $C_\gamma$ is a constant related to the maximum information gain, $d$ is the dimension of the domain $\mathcal{A}$, then when $t = T^*$, we obtain the $\zeta$-optimal objective function value $f(a_{opt})$, where $f(a^*) - f(a_{opt}) \leq \zeta$ with probability at least $1 - \delta$.*

Theorem 5.2 guarantees the convergence of the maximization stage, and $T^*$ counts the iteration numbers for this stage. The complete proof is presented in Appendix D.3.2.

## 6 EMPIRICAL STUDY

### 6.1 SIMULATIONS ON SYNTHETIC BENCHMARK FUNCTIONS

In this section, we evaluate the safe optimization performance of SAFECTRLBO against baseline methods using commonly used benchmark functions, including Camelback (2D), Hartmann (6D), and Gaussian (10D). We compare SAFECTRLBO with established safe BO methods such as SWARMSAFEOPT, SWARMSTAGEOPT (high-dimensional implementations of SAFEOPT and STAGEOPT), and the latest high-dimensional safe BO method, LINEBO (Kirschner et al., 2019). Additionally, we incorporate the state-of-the-art high-dimensional additive BO method, DUMBO (Bardou et al., 2024), as an unconstrained reference.

Since all three benchmark functions are designed for minimization, we invert their outputs and introduce safety constraints. For the Camelback function, the minimum value is approximately -1.0316, and after inversion, the maximum value becomes 1.0316. We set the safety constraint at 0, meaning that if an optimization method evaluates a point with a function value below 0, it constitutes a safety constraint violation. Similarly, for the Hartmann function, the inverted maximum value is approximately 3.32237, and we impose a safety constraint of 0.3. For the Gaussian 10D function, $f(x) = -exp(-4\|x\|_2^2)$, the inverted maximum value is 1, and we set the safety constraint at 0.1.

To ensure a fair comparison, we conducted 100 runs for each method on each benchmark function, plotting the mean and standard error of the simple regret (Figure 2). Each run begins with a randomly generated safe initial point (where the function value exceeds the safety threshold). All optimization methods were run for 150 iterations on the Camelback function and 200 iterations on the Hartmann and Gaussian functions. For LINEBO and DUMBO, we used publicly available implementations with default hyperparameters for each benchmark function. For SWARMSAFEOPT, SWARMSTAGEOPT, and SAFECTRLBO, we manually selected suitable hyperparameters. For detailed implementations, please refer to Appendix B.

**Results.** The results of simulations are presented in Figure 2. The state-of-the-art high-dimensional unconstrained BO method, DUMBO, achieved the lowest average simple regret across all benchmark functions. However, when safety constraints were considered, DUMBO violated the safety thresholds 1009, 3820, and 12,557 times over 100 runs on the Camelback, Hartmann, and Gaussian functions, respectively. In contrast, none of the safe BO algorithms violated any safety constraints under the settings defined in this section.

Due to the iteration limits of 150 and 200 for the benchmark functions—significantly fewer than the 400 to 1,000 iterations reported in Kirschner et al. (2019)—the three LINEBO variants did not perform as well. Instead, SWARMSTAGEOPT achieved lower simple regret than SWARMSAFEOPT, and SAFECTRLBO produced more competitive results.

Notably, the simple regret curves of LINEBO, SWARMSTAGEOPT, and SAFECTRLBO all exhibited segmented declines. LINEBO showed multiple segmented declines, while SWARMSTAGEOPT and SAFECTRLBO exhibited a two-stage decline. The multi-stage decline in LINEBO arises because it optimizes one subspace at a time; once a subspace converges, it moves to the next, leading to this trend. It benefits in finding the optimal solution but requires a substantial number of iterations. The two-stage decline in SWARMSTAGEOPT and SAFECTRLBO is due to the transition between the safe expansion stage and the maximization stage. This transition was set to occur after 15 iterations for the Camelback function and 50 iterations for the Hartmann and Gaussian functions, resulting in the observed segmented decline.

### 6.2 HARDWARE EXPERIMENTS ON A SPEEDGOAT REAL-TIME MACHINE

Compared to synthetic simulations, in real hardware experiments, running the same set of parameters twice under identical settings can lead to slight variations in results due to various real-world factors. This is particularly meaningful for testing the robustness of algorithms to random errors. In this section, hardware experiments are conducted using the SpeedGoat real-time machines, shown in Figure 3. The configuration includes a SpeedGoat controller with integrated speed and current loops, a SpeedGoat inverter, and a permanent magnet synchronous motor (PMSM). A schematic diagram of the entire configuration can be found in Appendix A.

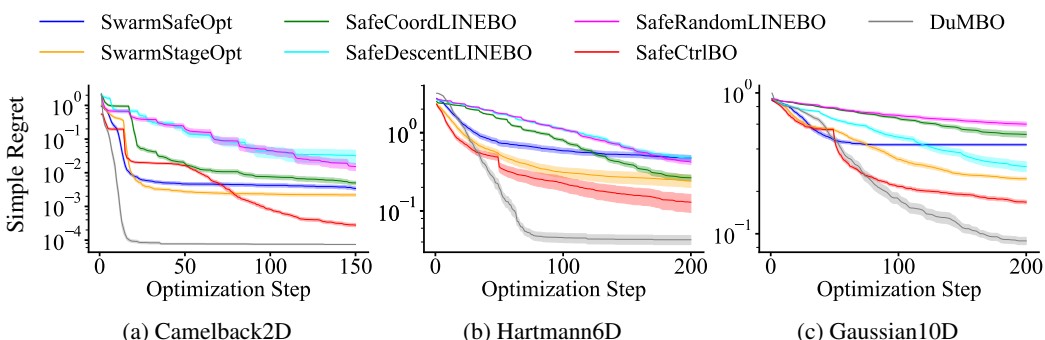

Figure 2: Optimization for synthetic benchmark functions.

The field-oriented control (FOC) algorithm within the SpeedGoat controller is used to control the PMSM. The FOC algorithm comprises a cascade control loop and is adjustable via MATLAB. The external controller is a speed controller responsible for regulating the motor's rotational speed. The internal controllers consist of two current controllers ($d$-axis and $q$-axis) that manage the current output from the inverter. These three controllers are interdependent, making it essential to adjust the six parameters across all controllers simultaneously to achieve the optimal parameter combination.

The objective is to determine the controller parameters that optimize the speed-tracking performance of the PMSM, aiming to maximize transient response speed while minimizing overshoot and steady-state error. This objective is crucial in various industrial applications, including precise robot joint control, industrial automation system control, and electric vehicle control. The transient response of the system is evaluated based on the 2% settling time, defined as the duration required for the response curve to reach and remain within 2% of the steady-state value. The performance function is then designed as:

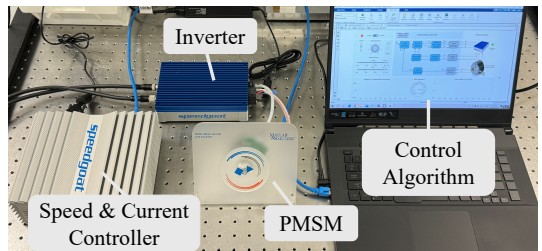

Figure 3: Hardware experimental setup.

$$J(t_s, O_s, e_{ss}) = w_s \cdot (t_0 - t_s) - w_o \cdot O_s - w_e \cdot e_{ss}, \tag{7}$$

where $w_s$, $w_o$, and $w_e$ are weight factors, $t_0$ is a time constant depending on the task, $t_s$ is the value of settling time, $O_s$ is the value of overshoot, and $e_{ss}$ is the value of steady-state error. The weight factors can be manually defined, and we choose $w_s = 20$, $w_o = 1.5$, $w_e = 4$, $t_0 = 2.5$ in the experiments.

To guarantee safety, the motor system must remain stable, so the steady-state error should be controlled within a narrow range. Additionally, the control signal must be moderated to prevent excessive current, which could potentially damage the motor hardware. To address these concerns, two safety functions have been designed, pertaining to the magnitude of the steady-state error and the amplitude of the control signal:

$$G_e = C_{e0} - w'_e \cdot e_{ss}, \quad G_u = C_{u0} - w_u \cdot \sum_{t=0}^{1} u(t)^2, \tag{8}$$

where $C_{e0}$ and $C_{u0}$ are constants defined according to the system characteristics, and $w'_e$ and $w_u$ are weight factors. We choose $C_{e0} = C_{u0} = 100$, $w'_e = 40$, and $w_u = 0.001$ in the experiments. The safety functions' minimum thresholds are set at $0$, indicating that any value below this threshold constitutes a violation of the safety constraints. The parameters predefined in the model serve as the initial settings, and evaluations of these initial settings against the safety functions indicate that their values meet this minimum threshold.

In the PMSM FOC control loop, the six controller parameters have different physical meanings and parameter ranges. Specifically, the proportional gain ($p$ gain) and integral gain ($i$ gain) of the speed

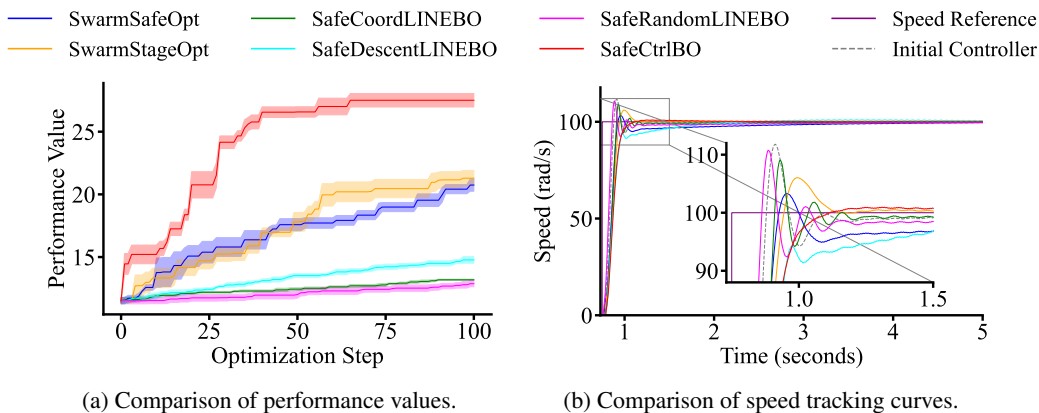

(a) Comparison of performance values.

(b) Comparison of speed tracking curves.

Figure 4: Hardware experiment results.

controller are set within the range $[0.01, 0.5]$, while the $p$ gains of the $d$-axis and $q$-axis current controllers are set within $[0.1, 1]$, and their $i$ gains are set within $[1, 200]$. The differing physical significance of these parameters leads to varying impacts on the performance and safety functions. The $p$ gain and $i$ gain of the speed controller and $q$-axis current controller significantly influence the settling time, overshoot, and steady-state error of the speed tracking curve, which greatly affects the performance function $J$. In contrast, the $p$ gain and $i$ gain of the $d$-axis current controller have minimal impact on speed tracking but influence current and flux, which substantially affects the safety function $G_u$. The additive kernel in SAFECTRLBO can adapt to this optimization problem by allowing users to adjust the signal variance and lengthscale of the base kernels, based on the smoothness of the objective functions within each parameter's range (the degree to which each parameter influences the performance and safety functions).

Due to hardware safety concerns, only safe BO algorithms are compared in this section, including SWARMSAFEOPT, SWARMSTAGEOPT, LINEBO, and SAFECTRLBO. Since LINEBO can only handle the performance function as the safety function, the $G_e$ and $G_u$ functions defined in this section are not evaluated in LINEBO. To ensure a fair comparison, we conducted 5 runs for each method, each consisting of 100 iterations. The mean and standard error of the performance curves for all methods are presented in Figure 4a. Additionally, the highest performance results from the 5 runs for each method are illustrated in Figure 4b, with the performance metrics detailed in Table 1.

**Results.** As shown in Figure 4a, all six methods could continuously optimize the PMSM speed tracking performance, with SAFECTRLBO showing the most significant improvements. From the curves in Figure 4b and the performance metrics in Table 1, the best result obtained by SAFEC-TRLBO had the smallest overshoot, the shortest $2\%$ settling time, and the second-smallest steady-state error. In terms of constraint violation, since the three LINEBO methods were unable to constrain the steady-state error and control signal, we only recorded their violations of the performance threshold. SAFECOORDINATELINEBO violated the performance threshold once, SAFEDESCENT-LINEBO violated it 4 times, and SAFERANDOMLINEBO violated it 5 times. The remaining methods considered both performance and safety thresholds. SWARMSAFEOPT violated thresholds 39 times across five runs, SWARMSTAGEOPT violated them 61 times, and SAFECTRLBO violated them 39 times. A more detailed discussion on constraint violation can be found in Appendix ???.

Notably, in this experiment, the SAFECTRLBO method employed a full additive kernel, where the additive kernels of all six orders were summed to form the kernel for the Gaussian process. While this enhanced optimization efficiency, it introduced a substantial computational cost. When using the potential expander set $\mathcal{E}_n$ defined in previous safe BO methods for safe exploration, each iteration took approximately 48 seconds. However, using the set of safe boundary points $\mathcal{B}_n$ reduced the iteration time to around 28 seconds. This highlights the significance of simplifying the potential expander set when performing safe BO with additive kernels.

Table 1: Performance comparison of the best PMSM speed tracking curves optimized by different methods. The best results are written in **bold** text, and the second-best results are underlined.

| Method | $J(a^*)\uparrow$ | $O_s\ (rad/s)\downarrow$ | $e_{ss}\ (rad/s)\downarrow$ | $2\%\ t_s\ (s)\downarrow$ |
|---|---|---|---|---|
| Initial Controller | 11.6788 | 11.789 | 0.067 | 0.301 |
| SWARMSAFEOPT | 21.4277 | 3.323 | 0.417 | 1.721 |
| SWARMSTAGEOPT | 23.7091 | 6.063 | **0.030** | 0.327 |
| SAFECOORDLINEBO | 13.2615 | 9.743 | 0.081 | 0.431 |
| SAFEDESCENTLINEBO | 14.734 | 1.028 | 1.681 | 0.988 |
| SAFERANDOMLINEBO | 12.2435 | 10.747 | 0.409 | 0.477 |
| SAFECTRLBO | **28.7893** | **0.956** | 0.037 | **0.284** |

## 7 CONCLUSION

In this study, we implemented additive Gaussian kernels to enhance the efficiency of safe Bayesian optimization in complex control problems, while also proposing a simplified safe expansion process to mitigate the additional computational cost introduced by the additive kernels. Empirical results from both benchmark function simulations and hardware experiments demonstrate that our method is highly competitive among baseline approaches and can be seamlessly integrated into real-world complex control applications. Although tested primarily for PMSM control, the proposed algorithm has the potential to be applied to other control architectures and to a wide range of robotic and mechatronic systems.

However, a limitation of this work is that the computation of high-dimensional additive kernels becomes increasingly complex as the dimensions of control problems grow. For instance, a 100-dimensional problem would involve 100 orders of additive kernels. Designing and combining these kernel components effectively requires extensive domain knowledge and substantial experimental effort. Furthermore, the computational cost associated with these combinations becomes a significant barrier to scaling this approach to very high-dimensional problems. A possible solution is to use kernel selection methods (Cristianini et al., 2001; Kandola et al., 2002; Ding et al., 2020) to obtain one or more additive kernels with the highest efficiency in exploring the parameter space, and use the selected kernels for subsequent optimization.

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

## A   ARCHITECTURE OF THE FIELD-ORIENTED CONTROL SCHEME ON A PMSM

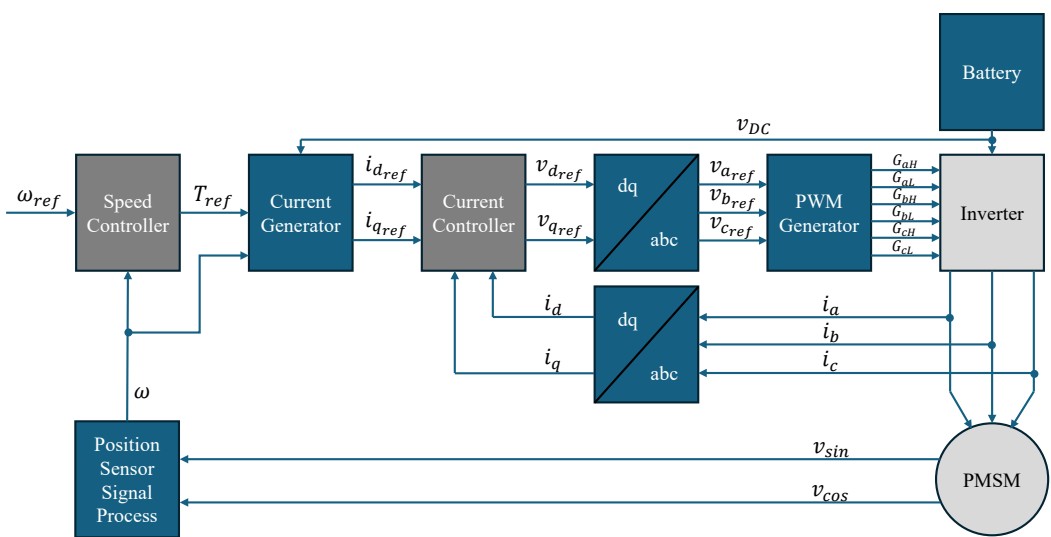

Figure 5: A simplified block diagram for PMSM FOC loops. The dark grey blocks represent controllers, and the light grey blocks represent plants.

## B   IMPLEMENTATION DETAILS OF EXPERIMENTS

The implementation details of the proposed method can be accessed via `https://drive.google.com/drive/folders/1PKunCvjenu3B8QQn-t5jttx_b4yF8wZE?usp=sharing`

We will explain the choices of hyperparameters used in the experiments as follows:

### B.1   CHOICE OF $T_0$

$T_0$ represents the number of iterations in the safe exploration stage. According to Theorem 5.1, $T_0$ is theoretically finite. However, in the practical control optimization tasks discussed in the paper, excessive exploration is unrealistic. For instance, in our hardware experiment, 100 iterations required approximately 5 hours, with most of the time spent on motor performance testing. In this case, we selected $T_0 = 15$, and our controller achieved stable and good performance after about 45 iterations.

### B.2   CHOICE OF $\beta$ AND KERNEL VARIANCES AND LENGTHSCALES

The selection of Gaussian Process (GP) hyperparameters in the paper was primarily aimed at aligning with the baseline experiments to enable a fair comparison. We compared six baseline algorithms in the paper, three of which were from Kirschner et al. (2019). To ensure consistency, we used the same hyperparameters where possible, including the choice of $\beta = 2$.

In the hardware experiments, certain field-oriented control domain knowledge informed the selection of the variance and lengthscale of the base kernels. Field-oriented control comprises three loops, with varying impacts on motor response: - The $P$-gain and $I$-gain of the speed loop have the greatest impact on motor response. - The $P$-gain and $I$-gain of the $q$-axis current loop have a moderate impact. - The $P$-gain and $I$-gain of the $i$-axis current loop, as the reference current remains at zero, have the smallest impact and primarily affect signal safety.

Accordingly, we set the variances of $k_1$ and $k_2$ to be higher, allowing the GP to model larger variations in the performance function along these dimensions. Conversely, the variances of $k_5$ and $k_6$ were set lower, reflecting their lesser contribution to the overall model. Since the optimization

results were already excellent under this configuration, we did not further fine-tune the lengthscales. However, if fine-tuning were necessary, we believe that relatively smaller lengthscales should be selected for $k_1$ and $k_2$, and larger lengthscales for $k_5$ and $k_6$. Additionally, functions such as "kernel.variance.set_prior()" or "kernel.lengthscale.set_prior()" could be used for real-time fine-tuning.

## C  DISCUSSION ON CONSTRAINT VIOLATION IN HARDWARE EXPERIMENTS

### C.1  DISCUSSION ON SAFETY GUARANTEE

In the hardware experiment of our method, there are in total 39 constraint violations in 5 runs. Most constraint violations pertain to the first constraint function, where the performance value is lower than the preset threshold. The first constraint is a soft constraint, and a violation indicates that the performance is below expectations, such as exhibiting a large overshoot or a slower transient response. Importantly, this violation does not imply that the system is unsafe. The constraint could be set to $-\infty$ to ensure no violations; however, this may reduce optimization efficiency.

A smaller number of violations involve the second constraint function, representing the steady-state error being lower than the preset threshold. Such a violation implies that the steady-state value of the motor speed deviates significantly from the desired setpoint, e.g., requiring the motor to stabilize at $100\,\mathrm{rad/s}$, but it instead stabilizes at $90\,\mathrm{rad/s}$. Similar to the first constraint, this violation does not indicate system unsafety. Instead, it is introduced to enhance the optimization efficiency of the algorithm by discouraging the selection of parameters that lead to high steady-state errors.

The third constraint ensures that the value of the function representing signal safety remains above the preset safe threshold throughout the optimization process, thereby guaranteeing the safety of the system during optimization.

### C.2  DISCUSSION ON TRADE-OFF BETWEEN SAFETY AND EFFICIENCY

The violations primarily occurred during the exploration phase. This can be reduced by adjusting the hyperparameters of the base kernels or modifying the performance function to make it smoother. However, such adjustments may require more iterations, thus increasing computational costs in practical applications, leading to a trade-off between safety and optimization efficiency.

Moreover, as noted by (Sui et al., 2018), safe BO aims to ensure safety with high probability rather than guaranteeing 100% safety. Therefore, safe BO methods based on confidence intervals inherently have a small probability of constraint violation, regardless of how sophisticated the hyperparameter design may be. For example, constraint violations are also reported in (Khosravi et al., 2023).

### C.3  DISCUSSION ON HOW TO HANDLE THE TRADE-OFF

How to manage this trade-off depends on the specific task requirements. It requires a comprehensive optimization of multiple aspects of the task. For instance:

**Iteration tolerance**: How many iterations can the task afford? In systems where each iteration runs quickly and a large number of iterations do not cause significant wear, a relatively conservative strategy can be adopted. This may reduce optimization efficiency but enhance safety guarantee.

**Safety requirements**: How stringent are the safety requirements? For example, the control optimization process of a drone system may have high safety requirements, as unsafe control parameters may result in crashes or collisions. In such cases, safety must be guaranteed even at the cost of reduced optimization efficiency. Conversely, in automotive motor control optimization, safety requirements are relatively lower. Some poorly performing control parameters may require human intervention to early terminate their operation, but in most cases, they will not cause direct damage. Therefore, as set in the hardware experiment, we can appropriately relax the constraints on tracking performance while ensuring signal safety to improve optimization efficiency.

# D DETAILED PROOFS

## D.1 PROOFS FOR THE PROPERTIES OF ADDITIVE GAUSSIAN KERNELS

**Lemma D.1.** *The RKHS $\mathcal{H}$ corresponding to the additive kernel $K$ composed of one-dimensional Gaussian kernels $K_i$ is a complete inner product space composed of the direct sum of the RKHSs corresponding to each one-dimensional Gaussian kernel, and the additive kernel $K$ is a positive definite kernel function, which conforms to the properties of the reproducing kernel.*

*Proof.* For one-dimensional inputs $x_i$ and $y_i$, the Gaussian kernel is defined as:

$$K_i(x_i, y_i) = \exp\left(-\frac{\|x_i - y_i\|^2}{2\sigma_i^2}\right).$$

Each one-dimensional Gaussian kernel $K_i$ has a corresponding RKHS, denoted by $\mathcal{H}_i$, which satisfies the reproducing property.

Suppose there are $d$ one-dimensional Gaussian kernels, then the additive kernel is constructed as:

$$K(x, y) = \sum_{i=1}^{d} K_i(x_i, y_i),$$

where $x = (x_1, x_2, \ldots, x_d)$ and $y = (y_1, y_2, \ldots, y_d)$.

We first prove that $K(x, y)$ is a positive definite kernel. For any sample points $\{x_1, x_2, \ldots, x_n\}$ and corresponding non-zero weight vector $\alpha = (\alpha_1, \alpha_2, \ldots, \alpha_n), \alpha \in \mathbb{R}^n$, there is:

$$\sum_{j=1}^{n}\sum_{k=1}^{n} \alpha_j \alpha_k K(x_j, x_k) = \sum_{j=1}^{n}\sum_{k=1}^{n} \alpha_j \alpha_k \sum_{i=1}^{d} K_i((x_j)_i, (x_k)_i).$$

Since each $K_i$ is positive definite,

$$\sum_{j=1}^{n}\sum_{k=1}^{n} \alpha_j \alpha_k K_i((x_j)_i, (x_k)_i) \geq 0,$$

thus:

$$\sum_{j=1}^{n}\sum_{k=1}^{n} \alpha_j \alpha_k K(x_j, x_k) = \sum_{i=1}^{d}\sum_{j=1}^{n}\sum_{k=1}^{n} \alpha_j \alpha_k K_i((x_j)_i, (x_k)_i) \geq 0,$$

which shows that $K(x, y)$ is a positive definite kernel.

Now we prove that the RKHS corresponding to the additive kernel can be constructed from the RKHSs of the individual one-dimensional Gaussian kernels.

Assume $\mathcal{H}_i$ is the RKHS corresponding to the kernel $K_i$. For any $f_i \in \mathcal{H}_i$, there exists a function $K_i(\cdot, x_i)$ that satisfies the reproducing property:

$$f_i(x_i) = \langle f_i, K_i(\cdot, x_i) \rangle_{\mathcal{H}_i}.$$

We construct the new function space $\mathcal{H}$ as the direct sum of these $\mathcal{H}_i$:

$$\mathcal{H} = \bigoplus_{i=1}^{d} \mathcal{H}_i,$$

and in this new space, any function $f \in \mathcal{H}$ can be represented as:

$$f(x) = \sum_{i=1}^{d} f_i(x_i),$$

where $f_i \in \mathcal{H}_i$.

We define the new inner product in $\mathcal{H}$ as:

$$\langle f, g \rangle_{\mathcal{H}} = \sum_{i=1}^{d} \langle f_i, g_i \rangle_{\mathcal{H}_i}.$$

The completeness of $\mathcal{H}$ under this inner product is ensured because each $\mathcal{H}_i$ is complete, and the completeness of the direct sum space depends on the completeness of its component spaces.

Finally, we prove that the new RKHS $\mathcal{H}$ satisfies the reproducing property.

In each $\mathcal{H}_i$, the reproducing property is expressed as:

$$f_i(x_i) = \langle f_i, K_i(\cdot, x_i) \rangle_{\mathcal{H}_i},$$

and we need to prove that for the new kernel function $K$, the reproducing property holds:

$$f(x) = \langle f, K(\cdot, x) \rangle_{\mathcal{H}}.$$

Note that the new kernel function $K$ can be expressed as:

$$K(x, y) = \sum_{i=1}^{d} K_i(x_i, y_i),$$

therefore, for $f \in \mathcal{H}$ and any $x \in \mathcal{X}$,

$$f(x) = \sum_{i=1}^{d} f_i(x_i) = \sum_{i=1}^{d} \langle f_i, K_i(\cdot, x_i) \rangle_{\mathcal{H}_i}.$$

Using the definition of the new inner product,

$$f(x) = \sum_{i=1}^{d} \langle f_i, K_i(\cdot, x_i) \rangle_{\mathcal{H}_i} = \langle f, K(\cdot, x) \rangle_{\mathcal{H}},$$

which shows that the new kernel function $K$ satisfies the reproducing property in the new RKHS $\mathcal{H}$.

Therefore, the RKHS $\mathcal{H}$ corresponding to the additive kernel $K$ is constructed from the direct sum of the RKHSs of the individual one-dimensional Gaussian kernels. The additive kernel $K$ is a positive definite kernel and satisfies the reproducing property in its RKHS. ∎

Lemma D.1 proves the existence of RKHS for the additive kernels composed of the one-dimensional Gaussian kernels. The main idea is to prove that the additive kernel $K$ is a positive definite kernel function, and its corresponding RKHS $\mathcal{H}$ has a complete inner product structure and satisfies the reproducing property.

**Theorem D.1.** *If the norm of an objective function $f$ is bounded by $B_i$ in each of the RKHSs corresponding to the one-dimensional Gaussian kernels $K_i$, $i \in \{1, 2, \ldots, d\}$, then the norm of $f$ is bounded by $B$ in the RKHS associated with the additive kernel $K$ composed of $K_i$, where $B = \sum_{i=1}^{d} B_i$.*

*Proof.* Assume there are $d$ one-dimensional Gaussian kernels $K_i$, each corresponding to an RKHS $\mathcal{H}_i$. The additive kernel $K$ is defined as:

$$K(x, y) = \sum_{i=1}^{d} K_i(x_i, y_i),$$

where $x = (x_1, x_2, \ldots, x_d)$ and $y = (y_1, y_2, \ldots, y_d)$.

If a function $f$ has bounded norms in each of the RKHSs corresponding to the one-dimensional Gaussian kernels, there are:

$$\|f_i\|_{k_i}^2 \leq B_i,$$

where $\|f_i\|_{k_i}^2$ denotes the norm of $f$ in the RKHS $\mathcal{H}_i$ corresponding to each one-dimensional Gaussian kernel.

The RKHS $\mathcal{H}$ of the additive kernel $K$ is the direct sum of the RKHSs $\mathcal{H}_i$:

$$\mathcal{H} = \bigoplus_{i=1}^{d} \mathcal{H}_i.$$

As defined in the proof of Lemma D.1, in the RKHS $\mathcal{H}$, any function $f$ can be represented as:

$$f(x) = \sum_{i=1}^{d} f_i(x_i),$$

where $f_i \in \mathcal{H}_i$, then the norm of a function $f$ in the new RKHS $\mathcal{H}$ can be defined as:

$$\|f\|_K^2 = \sum_{i=1}^{d} \|f_i\|_{k_i}^2.$$

Since the norm of $f$ in each one-dimensional RKHS $\mathcal{H}_i$ is bounded by $B_i$:

$$\|f_i\|_{k_i}^2 \leq B_i,$$

there is:

$$\|f\|_K^2 = \sum_{i=1}^{d} \|f_i\|_{k_i}^2 \leq \sum_{i=1}^{d} B_i.$$

Let $B = \sum_{i=1}^{d} B_i$, then:

$$\|f\|_K^2 \leq B,$$

which shows that $f$ has a bounded norm in the RKHS associated with the additive kernel $K$. ∎

Based on Lemma D.1, Theorem D.1 ensures the additive kernels satisfy the first assumption in safe BO. It is proved by demonstrating that the norm of $f$ in the RKHS $\mathcal{H}$ of the additive kernel is the sum of its norms in the individual RKHSs $\mathcal{H}_i$ of the one-dimensional Gaussian kernels, ensuring the overall boundedness.

**Theorem D.2.** *If all the one-dimensional Gaussian kernels $K_i$ that constitute the additive kernel $K$ are $L_i$-Lipschitz-continuous, then $K$ satisfies $L$-Lipschitz continuity, where $L = \left( \sum_{i=1}^{d} L_i \right) \sqrt{d}$.*

*Proof.* A Gaussian kernel is defined as:

$$K(x, y) = \exp \left( -\frac{\|x - y\|^2}{2\sigma^2} \right)$$

For the Lipschitz continuity of the Gaussian kernel, if we consider any two points $x$ and $y$ in the input space $\mathcal{X}$, we need to prove that there exists a constant $L$ such that:

$$|K(x, z) - K(y, z)| \leq L \|x - y\|$$

for all $z \in \mathcal{X}$.

Given that each one-dimensional Gaussian kernel $K_i(x_i, y_i) = \exp \left( -\frac{(x_i - y_i)^2}{2\sigma_i^2} \right)$ is $L_i$-Lipschitz-continuous, then there exists a constant $L_i$ such that:

$$|K_i(x_i, z_i) - K_i(y_i, z_i)| \leq L_i |x_i - y_i|$$

for all $x_i, y_i, z_i \in \mathcal{X}_i$.

To prove that the additive kernel $K(x, y) = \sum_{i=1}^{d} K_i(x_i, y_i)$ satisfies Lipschitz continuity, we need to show that there exists a constant $L$ such that:

$$|K(x, z) - K(y, z)| \leq L \|x - y\|$$

for all $x, y, z \in \mathcal{X}$.

Consider the difference of additive kernels:

$$|K(x,z) - K(y,z)| = \left| \sum_{i=1}^{d} K_i(x_i, z_i) - \sum_{i=1}^{d} K_i(y_i, z_i) \right|.$$

According to the triangle inequality,

$$\left| \sum_{i=1}^{d} K_i(x_i, z_i) - \sum_{i=1}^{d} K_i(y_i, z_i) \right| \leq \sum_{i=1}^{d} |K_i(x_i, z_i) - K_i(y_i, z_i)|.$$

Since each $K_i$ is $L_i$-Lipschitz-continuous,

$$|K_i(x_i, z_i) - K_i(y_i, z_i)| \leq L_i |x_i - y_i|,$$

thus:

$$|K(x,z) - K(y,z)| \leq \sum_{i=1}^{d} L_i |x_i - y_i|.$$

Let $\|x - y\|_1 = \sum_{i=1}^{d} |x_i - y_i|$ represents the $\ell_1$ norm of the vector, there is:

$$\sum_{i=1}^{d} L_i |x_i - y_i| = \left( \sum_{i=1}^{d} L_i \right) \|x - y\|_1.$$

Note that there is the following relationship between the $\ell_1$ norm and the $\ell_2$ norm:

$$\|x - y\|_1 \leq \sqrt{d} \|x - y\|,$$

thus:

$$\sum_{i=1}^{d} L_i |x_i - y_i| \leq \left( \sum_{i=1}^{d} L_i \right) \sqrt{d} \|x - y\|.$$

Let $L = \left( \sum_{i=1}^{d} L_i \right) \sqrt{d}$, then:

$$|K(x,z) - K(y,z)| \leq L \|x - y\|,$$

which shows that the additive kernel $K$ satisfies $L$-Lipschitz continuity. ∎

Theorem D.2 ensures the additive kernels satisfy the second assumption in safe BO. Given the properties of Lipschitz continuity for each $K_i$, Theorem D.2 is proved by demonstrating that the sum of these Lipschitz continuous functions, $K$, retains the Lipschitz property with a constant $L$, that is the sum of the individual $L_i$.

## D.2 PROOFS FOR SIMPLIFYING THE SAFE EXPANSION STAGE

**Lemma D.2.** *Along any straight line from $x_{oes}$ to $x_{sb}$, the distance $d_i(\lambda) = \|x(\lambda) - x_i\|$ to each evaluated safe point $x_i$ increases monotonically with $\lambda \in [0, 1]$, where:*

$$x(\lambda) = x_{oes} + \lambda(x_{sb} - x_{oes}).$$

*Proof.*

$$\frac{dd_i(\lambda)}{d\lambda} = \frac{1}{d_i(\lambda)} \left( x(\lambda) - x_i \right)^{\top} \left( x_{\mathrm{sb}} - x_{\mathrm{oes}} \right).$$

Since $x_{\mathrm{sb}}$ is farther from $x_i$ than $x_{\mathrm{oes}}$ is, the inner product $\left( x(\lambda) - x_i \right)^{\top} \left( x_{\mathrm{sb}} - x_{\mathrm{oes}} \right) \geq 0$.

Therefore,

$$\frac{dd_i(\lambda)}{d\lambda} \geq 0.$$

The distance $d_i(\lambda)$ increases monotonically with $\lambda$. ∎

**Theorem D.3.** *In a Gaussian Process with an RBF kernel in any dimension $D \geq 1$, when the outermost region $O_n$ is sufficiently large, the predictive variance $\sigma_t^2(x)$ increases monotonically along any straight line from an Outermost Evaluated Safe Point $x_{oes}$ to its nearest Safe Boundary Point $x_{sb}$. Consequently, the point with maximum predictive uncertainty in the expander set $E_t$ lies on the safe boundary.*

*Proof.* The predictive variance at a test point $x$ is given by:

$$\sigma_t^2(x) = k(x, x) - \mathbf{k}_t(x)^\top \mathbf{K}_t^{-1} \mathbf{k}_t(x),$$

where $k(x, x) = \sigma_f^2$, $\mathbf{k}_t(x) = [k(x, x_1), k(x, x_2), \ldots, k(x, x_n)]^\top$, and $\mathbf{K}_t$ is the $n \times n$ kernel matrix with entries $[\mathbf{K}_t]_{ij} = k(x_i, x_j)$.

We assume noise-free observations ($\sigma_n^2 = 0$) for simplicity. The RBF kernel depends on the squared Euclidean distance:

$$k(x, x_i) = \sigma_f^2 \exp\left(-\frac{\|x - x_i\|^2}{2l^2}\right).$$

We aim to show that $\sigma_t^2(x(\lambda))$ increases monotonically with $\lambda \in [0, 1]$.

The predictive variance along the path is:

$$\sigma_t^2(x(\lambda)) = \sigma_f^2 - \mathbf{k}_t(x(\lambda))^\top \mathbf{K}_t^{-1} \mathbf{k}_t(x(\lambda)).$$

Take the derivative of $\sigma_t^2(x(\lambda))$:

$$\frac{d\sigma_t^2(x(\lambda))}{d\lambda} = -2 \left(\frac{d\mathbf{k}_t(x(\lambda))}{d\lambda}\right)^\top \mathbf{K}_t^{-1} \mathbf{k}_t(x(\lambda)).$$

Compute $\dfrac{d\mathbf{k}_t(x(\lambda))}{d\lambda}$:

$$\frac{d\mathbf{k}_t(x(\lambda))}{d\lambda} = -\frac{1}{l^2} \left[(x(\lambda) - x_1)^\top (x_{\text{sb}} - x_{\text{oes}}) k(x(\lambda), x_1), \ldots, (x(\lambda) - x_n)^\top (x_{\text{sb}} - x_{\text{oes}}) k(x(\lambda), x_n)\right]^\top.$$

When $O_n$ is sufficiently large, since the observation obtained by safe exploration is a collection of clustered points around the initial safe prior, we can reasonably assume that the values of different $x(\lambda) - x_i$ are close to the same ($x_i$ are the evaluated safe points around the initial safe prior). We have:

$$\frac{d\sigma_t^2(x(\lambda))}{d\lambda} = \frac{2}{l^2} (x(\lambda) - x_i)^\top (x_{\text{sb}} - x_{\text{oes}}) \mathbf{k}_t(x(\lambda))^\top \mathbf{K}_t^{-1} \mathbf{k}_t(x(\lambda)),$$

Since $(x(\lambda) - x_i)^\top (x_{\text{sb}} - x_{\text{oes}}) \geq 0$, there is:

$$\frac{d\sigma_t^2(x(\lambda))}{d\lambda} \geq 0.$$

The predictive variance $\sigma_t^2(x(\lambda))$ increases monotonically with $\lambda$. ∎

*Proof of Theorem 4.1.* Since $\sigma_t^2(x(\lambda))$ increases monotonically along the path from $x_{\text{oes}}$ to $x_{\text{sb}}$, it attains its maximum at $x_{\text{sb}}$.

Therefore, when $O_n$ is sufficiently large, the point with maximum predictive variance in $O_n$ lies on the set of safe boundary points $\mathcal{B}_n$. ∎

**Visualization in 1D and 2D cases.** We show the effectiveness of simplifying the safe expansion stage visually in 1D and 2D. We use SAFEOPT as the baseline algorithm and replace the potential expander set $\mathcal{E}_n$ used in SAFEOPT with the proposed set of safe boundary points $\mathcal{B}_n$ as the comparison algorithm. Safe optimization was performed on 1D and 2D simulation functions with the safety threshold set to 0. As shown in Figure 6 and 7, the comparison algorithm using $\mathcal{B}_n$ can acquire the same prediction points as SAFEOPT.

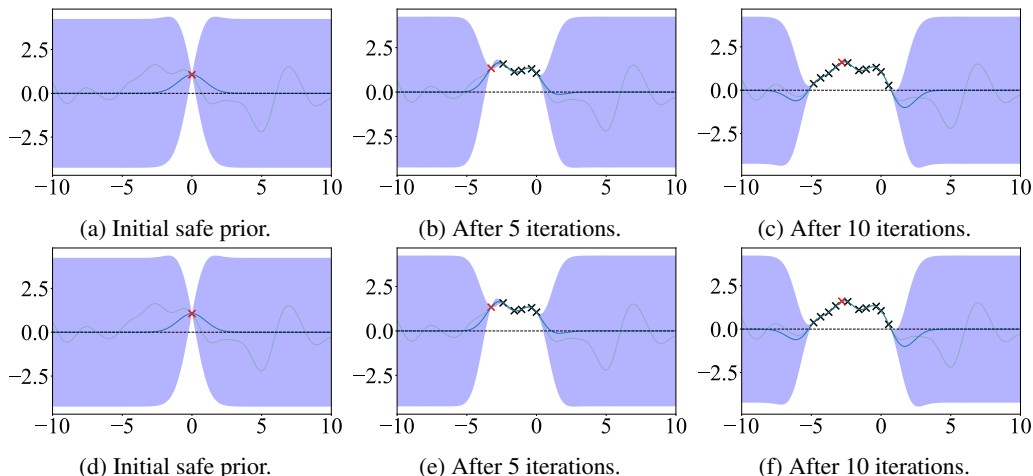

Figure 6: 1D visualization for the safe optimization process. The blue curve and purple shading represent the mean and confidence interval of the Gaussian process, respectively. The green curve represents the true value of the unknown function. Red markers indicate the prediction points acquired in the current iteration, and black markers show the previous prediction points. The black dashed line represents the safety threshold, which we set to 0. (a) - (c) are the results using the potential expander set $\mathcal{E}_n$, and (d) - (f) are the results using the set of safe boundary points $\mathcal{B}_n$.

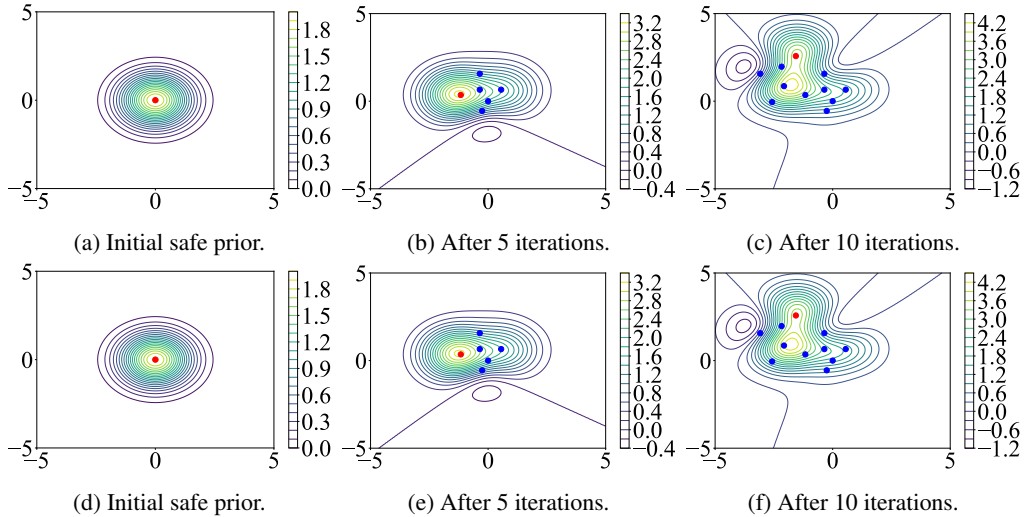

Figure 7: 2D visualization for safe exploration. The contour values represent the predicted values of the unknown function by the Gaussian process. Red markers indicate the prediction points acquired in the current iteration, while blue markers represent the previous prediction points. (a) - (c) are the results using the potential expander set $\mathcal{E}_n$, and (d) - (f) are the results using the set of safe boundary points $\mathcal{B}_n$.

### D.3 PROOFS FOR THEORETICAL RESULTS

#### D.3.1 PROOF OF THEOREM 5.1

**Lemma D.3.** *The predictive variance at a point $x$ after observing a point $x_i$ at distances $r_j = |x_j - x_{i,j}|$ in each dimension $j$ is bounded by:*

$$\sigma_t^2(x) \le \sum_{j=1}^{d} \sigma_f^2 \left( 1 - \exp\left( -\frac{r_j^2}{l_j^2} \right) \right).$$

*Proof.* For an RBF kernel

$$k(x, x') = \sigma_f^2 \exp\left(-\frac{\|x - x'\|^2}{2l^2}\right),$$

when we have observations at $x_i$, the predictive variance at a new point $x$ is:

$$\sigma_t^2(x) = \sigma_f^2 - \mathbf{k}_t(x)^\top \left(\mathbf{K}_t + \sigma_n^2 \mathbf{I}\right)^{-1} \mathbf{k}_t(x).$$

Assuming zero noise ($\sigma_n^2 = 0$) for simplicity, we have:

$$\sigma_t^2(x) = \sigma_f^2 - \frac{k(x, x_i)^2}{k(x_i, x_i)}.$$

Since $k(x_i, x_i) = \sigma_f^2$, this simplifies to:

$$\sigma_t^2(x) = \sigma_f^2 - \frac{k(x, x_i)^2}{\sigma_f^2}$$

$$= \sigma_f^2 - \frac{\left(\sigma_f^2 \exp\left(-\frac{r^2}{2l^2}\right)\right)^2}{\sigma_f^2}$$

$$= \sigma_f^2 \left(1 - \exp\left(-\frac{r^2}{l^2}\right)\right).$$

Similarly, for the additive Gaussian kernel

$$k_{\text{add}}(x, x') = \sum_{j=1}^d \sigma_f^2 \exp\left(-\frac{(x_j - x'_j)^2}{2l_j^2}\right),$$

assuming we have observations at $x_i$, the predictive variance at a new point $x$ is:

$$\sigma_t^2(x) = k_{\text{add}}(x, x) - \frac{k_{\text{add}}(x, x_i)^2}{k_{\text{add}}(x_i, x_i)},$$

where $k_{\text{add}}(x, x) = \sum_{j=1}^d \sigma_f^2 = d\sigma_f^2$, and we have:

$$k_{\text{add}}(x, x_i) = \sum_{j=1}^d \sigma_f^2 \exp\left(-\frac{(x_j - x_{i,j})^2}{2l_j^2}\right).$$

Thus,

$$\sigma_t^2(x) = d\sigma_f^2 - \frac{\left(\sum_{j=1}^d \sigma_f^2 \exp\left(-\frac{(x_j - x_{i,j})^2}{2l_j^2}\right)\right)^2}{d\sigma_f^2}.$$

Simplifying, we have:

$$\sigma_t^2(x) = d\sigma_f^2 - \frac{\sum_{j=1}^d \left(\sigma_f^2 \exp\left(-\frac{(x_j - x_{i,j})^2}{l_j^2}\right)\right) + \sum_{j \neq k} \sigma_f^2 \exp\left(-\frac{(x_j - x_{i,j})^2 + (x_k - x_{i,k})^2}{l_j^2 + l_k^2}\right)}{d\sigma_f^2}.$$

We can bound the variance by considering only the diagonal terms. Thus:

$$\sigma_t^2(x) \leq \sum_{j=1}^d \sigma_f^2 \left(1 - \exp\left(-\frac{(x_j - x_{i,j})^2}{l_j^2}\right)\right)$$

$$= \sum_{j=1}^d \sigma_f^2 \left(1 - \exp\left(-\frac{r_j^2}{l_j^2}\right)\right).$$

$\blacksquare$

**Lemma D.4.** *To include a point $x$ in the $\epsilon$-reachable safe region, it suffices that the predictive variance satisfies:*

$$\sigma_{t-1}^{(i)}(x) \leq \frac{\epsilon}{2\beta_t}.$$

*Proof.* In the $\epsilon$-reachable safe region, $\forall i, max_{a \in \mathcal{B}_t} 2\beta_t \sigma_{t-1}^{(i)}(x) \leq \epsilon$. Thus, $\sigma_{t-1}^{(i)}(x) \leq \frac{\epsilon}{2\beta_t}$. ∎

**Lemma D.5.** *The $\epsilon$-reachable safe region $\mathcal{R}_\epsilon$ can be covered by $N$ hypercubes of side length $r_j$, where:*

$$r_j = \frac{l_j \epsilon}{2\sqrt{d}\sigma_f \beta_t},$$

*and*

$$N = \prod_{j=1}^{d} \left\lceil \frac{L_j}{r_j} \right\rceil,$$

*with $L_j$ being the length of the domain in dimension $j$.*

*Proof.* From Lemma D.4, we need to ensure:

$$\sigma_{t-1}(x) = \left( \sum_{j=1}^{d} \sigma_{t-1,j}^2(x_j) \right)^{1/2} \leq \frac{\epsilon}{2\beta_t}.$$

To satisfy this inequality, it suffices that each term satisfies:

$$\sigma_{t-1,j}(x_j) \leq \frac{\epsilon}{2\beta_t\sqrt{d}}.$$

From Lemma D.3, the predictive variance in dimension $j$ after observing at $x_{i,j}$ is bounded by:

$$\sigma_{t-1,j}^2(x_j) \leq \sigma_f^2 \left( 1 - \exp\left( -\frac{(x_j - x_{i,j})^2}{l_j^2} \right) \right).$$

It thus suffices that:

$$\sigma_f \sqrt{1 - \exp\left( -\frac{(x_j - x_{i,j})^2}{l_j^2} \right)} \leq \frac{\epsilon}{2\beta_t\sqrt{d}}.$$

Solving for $(x_j - x_{i,j})^2$:

$$1 - \exp\left( -\frac{(x_j - x_{i,j})^2}{l_j^2} \right) \leq \left( \frac{\epsilon}{2\sigma_f \beta_t \sqrt{d}} \right)^2$$

$$\exp\left( -\frac{(x_j - x_{i,j})^2}{l_j^2} \right) \geq 1 - \left( \frac{\epsilon}{2\sigma_f \beta_t \sqrt{d}} \right)^2$$

$$-\frac{(x_j - x_{i,j})^2}{l_j^2} \geq \ln\left( 1 - \left( \frac{\epsilon}{2\sigma_f \beta_t \sqrt{d}} \right)^2 \right).$$

As $\epsilon$ is a very small positive real number, $\left( \frac{\epsilon}{2\sigma_f \beta_t \sqrt{d}} \right)^2$ is small, we can use $\ln(1 - x) \leq -x$, so it suffices that:

$$-\frac{(x_j - x_{i,j})^2}{l_j^2} \geq -\left( \frac{\epsilon}{2\sigma_f \beta_t \sqrt{d}} \right)^2 \implies (x_j - x_{i,j})^2 \leq \frac{l_j^2 \epsilon^2}{4\sigma_f^2 \beta_t^2 d}.$$

Thus,

$$r_j = |x_j - x_{i,j}| \leq \frac{l_j \epsilon}{2\sigma_f \beta_t \sqrt{d}}.$$

To cover the domain in dimension $j$ of length $L_j$, we need:

$$N_j = \left\lceil \frac{L_j}{r_j} \right\rceil.$$

Therefore, the total number of hypercubes is:

$$N = \prod_{j=1}^{d} N_j = \prod_{j=1}^{d} \left\lceil \frac{2\sigma_f \beta_t \sqrt{d} L_j}{l_j \epsilon} \right\rceil.$$

∎

*Proof of Theorem 5.1.* We aim to find a lower bound on $t^*$ for the safe expansion stage to converge to the $\epsilon$-reachable safe region $\mathcal{R}_\epsilon$ for all $t \geq t^*$.

From Lemma D.5, the number of observations needed is:

$$t^* \geq N = \prod_{j=1}^{d} \left\lceil \frac{2\sigma_f \beta_t \sqrt{d} L_j}{l_j \epsilon} \right\rceil.$$

Let:

$$C = \prod_{j=1}^{d} \left( \frac{2\sigma_f L_j}{l_j} \right),$$

$C$ is a constant depending on the size of $\mathcal{R}_\epsilon$ and kernel parameters.

Then we can simplify:

$$t^* \geq C \left( \frac{\beta_t \sqrt{d}}{\epsilon} \right)^d.$$

We define $\beta_t$ to have the form in Sui et al. (2018); Chowdhury & Gopalan (2017):

$$\beta_t = B + R\sqrt{2 \left( \gamma_{t-1} + 1 + \ln\left( \frac{1}{\delta} \right) \right)}.$$

For additive Gaussian kernels, the maximum information gain $\gamma_t$ scales as:

$$\gamma_t \leq \tilde{\gamma} d \ln t,$$

where $\tilde{\gamma}$ is a constant depending on the kernel and domain.

Therefore,

$$\beta_t \leq B + R\sqrt{2 \left( \tilde{\gamma} d \ln t + 1 + \ln\left( \frac{1}{\delta} \right) \right)}.$$

This shows $\beta_t$ is finite, then $t^*$ is finite and depends polynomially on $d$ and $1/\epsilon$. ∎

### D.3.2 PROOF OF THEOREM 5.2

**Lemma D.6.** *The instantaneous regret $r_t = f(x^*) - f(x_t)$ is bounded by:*

$$r_t \leq 2\beta_t \sigma_{t-1}(x_t).$$

*Proof.* From the safe BO algorithms, we have the upper confidence bound:

$$u_t(x) = \mu_{t-1}(x) + \beta_t \sigma_{t-1}(x)$$

and the lower confidence bound:

$$l_t(x) = \mu_{t-1}(x) - \beta_t \sigma_{t-1}(x).$$

With high probability $(1 - \delta)$, the true function value satisfies:

$$l_t(x) \leq f(x) \leq u_t(x), \quad \forall x \in \mathcal{D}, \forall t.$$

At iteration $t$:

- According to GP-UCB, $x_t = \arg\max_{x \in S_t} u_t(x)$, we have $u_t(x_t) \geq u_t(x^*) \geq f(x^*)$.

- The regret is:

$$r_t = f(x^*) - f(x_t) \leq u_t(x_t) - f(x_t) \leq u_t(x_t) - l_t(x_t) = 2\beta_t \sigma_{t-1}(x_t).$$

■

**Lemma D.7.** *The average predictive variance at $x_t$ satisfies:*

$$\sigma_{t-1}^2(x_t) \leq \frac{\gamma_{t-1}}{t-1}.$$

*Proof.* The total information gain up to iteration $t - 1$ is:

$$I(\mathbf{f}_{t-1}; \mathbf{y}_{t-1}) = \frac{1}{2} \sum_{i=1}^{t-1} \ln\left(1 + \frac{\sigma_{i-1}^2(x_i)}{\sigma_n^2}\right) \leq \gamma_{t-1}.$$

We assume $\sigma_n^2 \to 0$ for simplicity, then we have:

$$\sum_{i=1}^{t-1} \sigma_{i-1}^2(x_i) \leq 2\gamma_{t-1}.$$

Then for the average predictive variance, we get:

$$\sigma_{t-1}^2(x_t) \leq \frac{2\gamma_{t-1}}{t-1}.$$

■

*Proof of Theorem 5.2.* From Lemma C.6, the regret is bounded by:

$$r_t \leq 2\beta_t \sigma_{t-1}(x_t).$$

To achieve $r_t \leq \zeta$, it suffices that:

$$2\beta_t \sigma_{t-1}(x_t) \leq \zeta \implies \sigma_{t-1}(x_t) \leq \frac{\zeta}{2\beta_t}.$$

From Lemma D.7:

$$\sigma_{t-1}^2(x_t) \leq \frac{2\gamma_{t-1}}{t-1}.$$

Combining the two, it suffices that:

$$\frac{2\gamma_{t-1}}{t-1} \leq \left(\frac{\zeta}{2\beta_t}\right)^2.$$

From the definition of $\beta_t$:

$$\beta_t = B + R\sqrt{2\left(\gamma_{t-1} + 1 + \ln\left(\frac{1}{\delta}\right)\right)}.$$

Let $C_\beta = 1 + \ln\left(\frac{1}{\delta}\right)$, substitute $\beta_t$:

$$\frac{2\gamma_{t-1}}{t-1} \leq \left(\frac{\zeta}{2\left[B + R\sqrt{2\left(\gamma_{t-1} + C_\beta\right)}\right]}\right)^2.$$

Let $t = T^*$. Rearranged:

$$\frac{\gamma_{T^*-1}}{T^*-1}\left[B + R\sqrt{2\left(\gamma_{T^*-1} + C_\beta\right)}\right]^2 \leq \frac{\zeta^2}{4}.$$

For large $T^*$, $T^* - 1 \approx T^*$ and $\gamma_{T^*-1} \approx \gamma_{T^*}$.

For additive Gaussian kernels, the maximum information gain $\gamma_t$ scales logarithmically with $t$ and linearly with $d$:

$$\gamma_{T^*} \leq C_\gamma d \ln T^*,$$

Substitute back:

$$\frac{C_\gamma d \ln T^*}{T^*}\left[B + R\sqrt{2\left(C_\gamma d \ln T^* + C_\beta\right)}\right]^2 \leq \frac{\zeta^2}{4}.$$

∎

## E ABLATION STUDY FOR SIMPLIFYING THE SAFE EXPANSION STAGE

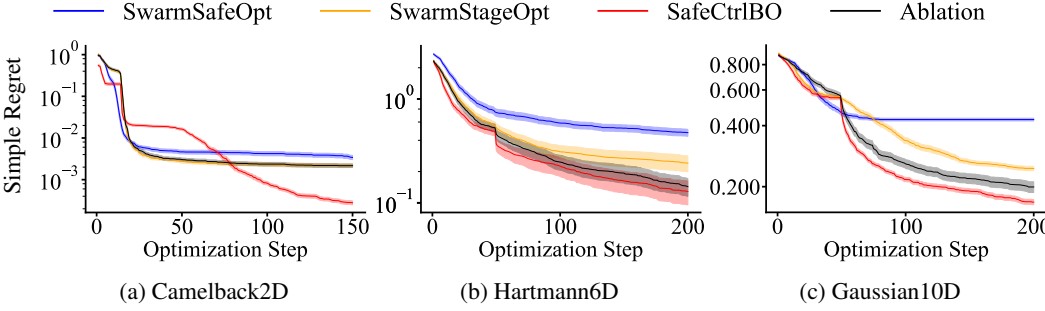

Figure 8: Ablation study using the synthetic benchmark functions.

In SAFECTRLBO, besides replacing Gaussian kernels with additive kernels, we also simplified the potential expander set $\mathcal{E}_n$ to the set of safe boundary points $\mathcal{B}_n$. In this section, we test whether the simplified safe exploration process results in any performance loss compared to previous safe BO methods.

Figure 8 shows the optimization results of different algorithms on the three benchmark functions, with the black curve representing the Ablation algorithm, which uses $\mathcal{B}_n$ for safe exploration but does not employ additive kernels. Therefore, the only difference between the Ablation algorithm and SWARMSTAGEOPT lies in the iteration strategy for safe exploration. As shown in Figure 8, the Ablation algorithm's results on all three benchmark functions are not inferior to those of SWARM-SAFEOPT and SWARMSTAGEOPT, suggesting that using the simplified safe exploration process does not lead to performance losses compared to using the potential expander set $\mathcal{E}_n$.

Moreover, in the Hartmann6D and Gaussian10D function experiments, the Ablation algorithm performs better than SWARMSTAGEOPT. The most plausible explanation is that, in high-dimensional settings or during the later stages of safe exploration, the acquisition function used in SAFECTRLBO exhibits higher safe exploration efficiency. This behavior is related to the assumption of Theorem D.3. If $O_n$ does not satisfy the condition of being sufficiently large, SAFECTRLBO and SWARM-STAGEOPT will make different predictions during the exploration stage.

# F  ADDITIONAL EXPERIMENTS ON NOISY BENCHMARK ENVIRONMENTS

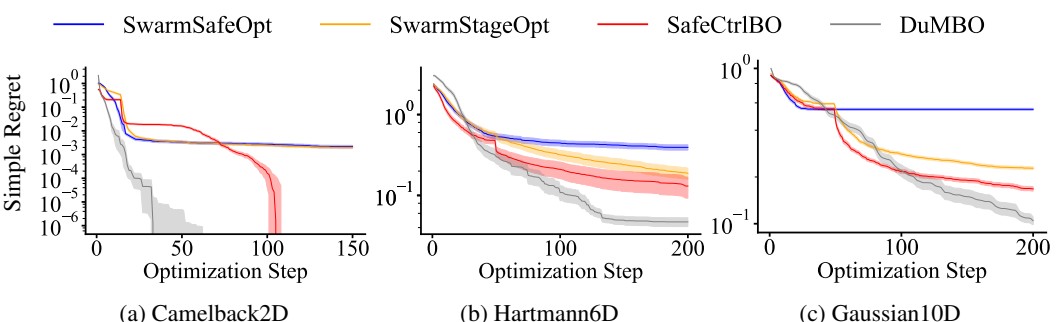

(a) Camelback2D      (b) Hartmann6D      (c) Gaussian10D

Figure 9: Simulations using noisy synthetic benchmark functions.

We added observation noise to the synthetic function environments and re-run the experiments. For the Camelback2D function, Gaussian noise with a standard deviation of 0.0001 was added, and the optimization results are presented in Figure 9a. Since SAFECTRLBO and DUMBO demonstrate highly effective performance, the true simple regret approaches zero after 150 iterations under the noise-free environment. Under noisy observations, this causes the simple regret to become negative.

Gaussian noise with a standard deviation of 0.001 was added for the Hartmann6D and Gaussian10D functions. The optimization results, shown in Figures 9b and 9c, exhibit minimal differences compared to Figure 2b and 2c.

