# OpenReview forum: "Safe Bayesian Optimization for Complex Control Systems via Additive Gaussian Processes"
_ICLR.cc/2025/Conference — Submitted to ICLR 2025_

### Official Review · Reviewer_zLs8 · 2024-10-21

**Soundness:** 4
**Presentation:** 4
**Contribution:** 4
**Rating:** 10
**Confidence:** 2

**Summary:**

The paper describes a method for safe Bayesian optimization suitable for control applications where safety concerns limit the set of parameters that can be tried. The main contribution is the use of additive kernels in safe BO, resulting in faster optimization. This contribution is of major significance when applying BO to physical systems, where control trials are slow and costly to perform. A second contribution is the speedup of the computation of the expander set, affecting the computational time of the algorithm. The proposed method has been verified on benchmark functions in simulation, as well as on a challenging control problem involving a physical set-up consisting of nested controllers regulating the velocity of a permanent magnet synchronous motor under field-oriented control.

**Strengths:**

The proposed method for using additive kernels in safety-aware BO leads to optimization in fewer trials, which is of great practical significance for physical control systems with safety limitations. The paper is written clearly, references prior work in the field, and explains very well the connections of the proposed method to that work. Although safe BO and additive functions are not novel ideas individually, their combination is original and based on significant technical insight. Properties of the proposed additive kernels are analyzed well theoretically. A novel acquisition function is proposed that is faster to compute than previously proposed ones.

Furthermore, the method is tested on a physical control system set-up that is of real practical use. Because the optimized parameters of the controllers are their proportional and integral gains, the proposed method could potentially find widespread use in industrial practice, where the use of PI controllers is very common and their tuning is known to be notoriously tricky and laborious.

**Weaknesses:**

The paper is somewhat incremental in the line of research on applying BO to safety-constrained control systems, and combines known ideas. Nevertheless, this combination required a significant technical insight, so it is far from obvious or trivial.

The experimental results on the physical system resulted in a significant number of constraint violations (39 for the method proposed by the authors). It is not clear what the consequences of these violations are in practice.

**Questions:**

What are the consequences of violations of constraints in this test application? Do they make the algorithm unsafe for use in practice? Is there a parameter that can be changed to reduce the number of violations? Does that result in a trade-off between performance and number of constraint violations, and if yes, how can this trade-off be handled?

---

> ### Author Response · Authors · 2024-11-20
>
> We thank the reviewer for the detailed review and are very grateful for the kind appreciation of the paper. We address the weaknesses raised by the reviewer and answer the reviewer's questions below. The corresponding modifications in the paper have been highlighted in cyan.
>
> ---
>
> ### **Comment 1**
>
> > “The experimental results on the physical system resulted in a significant number of constraint violations (39 for the method proposed by the authors). What are the consequences of violations of constraints in this test application? Do they make the algorithm unsafe for use in practice?”
>
> **Response:**
> In the hardware experiment of our method, there are in total 39 constraint violations in 5 runs. From the implementation details provided in the "SpeedGoat\_RT\_AdditiveBO\_exp.ipynb" file, we observe that most constraint violations pertain to the first constraint function, where the performance value is lower than the preset threshold. The first constraint is a soft constraint, and a violation indicates that the performance is below expectations, such as exhibiting a large overshoot or a slower transient response. Importantly, this violation does not imply that the system is unsafe. The constraint could be set to $ -\infty $ to ensure no violations; however, this may result in reduced optimization efficiency.
>
> A smaller number of violations involve the second constraint function, which represents the steady-state error being lower than the preset threshold. Such a violation implies that the steady-state value of the motor speed deviates significantly from the desired setpoint, e.g., requiring the motor to stabilize at $ 100 \text{rad/s} $, but it instead stabilizes at $ 90 \text{rad/s} $. Similar to the first constraint, this violation does not indicate system unsafety. Instead, it is introduced to enhance the optimization efficiency of the algorithm by discouraging the selection of parameters that lead to high steady-state errors.
>
> The third constraint ensures that the value of the function representing signal safety remains above the preset safe threshold throughout the optimization process, thereby guaranteeing the safety of the system during optimization.
>
> ---
>
> ### **Comment 2**
>
> > “Is there a parameter that can be changed to reduce the number of violations? Does that result in a trade-off between performance and number of constraint violations?”
>
> **Response:**
> Yes, there are parameters that can be adjusted. The choice of kernel hyperparameters and the weight factors in equations (7) and (8) will influence the number of constraint violations. By adjusting the weight factors in equations (7) and (8), we can control the smoothness of the performance function and constraint functions. Similarly, the variance and lengthscale of the base kernels allow us to adjust the smoothness of the function generated by the Gaussian Process. Ensuring appropriate smoothness for the performance function and constraint functions can satisfy the conditions of Theorems 5.1 and 5.2, leading to a high probability (not 100%, as similar analyses are made in [1], and constraint violations are reported in [2]) that the optimization process avoids violating the constraints.
>
> However, conservative settings will inevitably reduce optimization efficiency, representing the trade-off mentioned by the reviewer. As noted in the implementation details ("SafeCtrlBO\_Hartmann6D.ipynb"), using only the first-order additive kernel leads to more aggressive iterations, achieving the highest optimization efficiency but potentially violating constraints. Conversely, using only the highest-order additive kernel reduces optimization efficiency but ensures that the constraints are not violated with high probability. It is also possible to optimize kernel hyperparameters to ensure that all kernels avoid constraint violations with high probability. However, due to the trade-off, this adjustment also results in reduced optimization efficiency.
>
> The synthetic function benchmark results further corroborate this: the DuMBO algorithm exhibits the highest optimization efficiency among compared methods, but it violates constraints frequently, aligning with our conjecture.
>
> ---
>
> ### **References**
>
> 1. Yanan Sui, Vincent Zhuang, Joel Burdick, and Yisong Yue. Stagewise safe bayesian optimization with gaussian processes. In *Proc. of the 35th International Conference on Machine Learning (ICML)*, pp. 4781–4789, Stockholm, Sweden, 2018.
> 2. Mohammad Khosravi, Christopher Konig, Markus Maier, Roy S. Smith, John Lygeros, and Alisa Rupenyan. Safety-aware cascade controller tuning using constrained bayesian optimization. *IEEE Transactions on Industrial Electronics*, 70(2):2128–2138, 2023.

---

> > ### Comment · Reviewer_zLs8 · 2024-11-25
> > **Constraint violations explained**
> >
> > Thank you for explaining the nature of the constraint violations resulting from the application of the method. It appears that they are not safety related, but only failures to reach desired performance targets. This explanation is likely to increase the appeal of the method to practitioners who otherwise might be scared away by the phrase "constraint violation". I maintain my rating for the paper and recommend acceptance.

---

> > > ### Author Response · Authors · 2024-11-26
> > >
> > > Thank you very much for your response, and we sincerely appreciate your comments on improving the comprehensibility of the experimental section of this paper.

---

> ### Author Response · Authors · 2024-11-20
>
> ### **Comment 3**
>
> > “How can this trade-off be handled?”
>
> **Response:**
> How to manage this trade-off depends on the specific task requirements. It requires a comprehensive optimization of multiple aspects of the task. For instance:
> - **Iteration tolerance**: How many iterations can the task afford? In systems where each iteration runs quickly and a large number of iterations do not cause significant wear, a relatively conservative strategy can be adopted. This reduces optimization efficiency but enhances safety guarantees.
> - **Safety requirements**: How stringent are the safety requirements? For example, the control optimization process of a drone system may have high safety requirements, as unsafe control parameters may result in crashes or collisions. In such cases, safety must be guaranteed even at the cost of reduced optimization efficiency. Conversely, in automotive motor control optimization, safety requirements are relatively lower. Some poorly performing control parameters may require human intervention to early terminate their operation, but in most cases, they will not cause direct damage. Therefore, as set in the hardware experiment, we can appropriately relax the constraints on tracking performance while ensuring signal safety to improve optimization efficiency.
>
> ---
>
> We thank the reviewer for raising these questions, which helped us identify our problem in the explanation of the experimental results section. We have revised and improved it accordingly, and the modifications have been highlighted in cyan in the updated PDF. We hope these changes enhance the clarity and comprehensibility of the experimental results. If any issues or questions remain, please feel free to bring them up. We would be happy to provide further clarifications.

---

### Official Review · Reviewer_QXk1 · 2024-11-01

**Soundness:** 2
**Presentation:** 2
**Contribution:** 2
**Rating:** 1
**Confidence:** 4

**Summary:**

The paper addresses safe Bayesian optimization (BO) to optimize the parameters of cascaded control systems. To make safe BO more suitable for this task, additive kernels are used as a model, and a new definition of expander sets is introduced, which makes the BO optimization more compute efficient. The method is benchmarked on synthetic functions and a hardware setup, tuning parameters for a field-oriented control algorithm in a permanent magnet synchronous motor. Although the proposed method outperforms other benchmarks, all tested safe BO methods result in safety constraint violations.

**Strengths:**

(S1) Clear motivation and objective

(S2) Demonstration and benchmarking of  different safe BO algorithms on synthetic function and in a real-world use case. The application of multiple algorithms as baselines makes the results significant for the BO community.

(S3) The provided implementation gives clear insights into the method and the experiments.

In general, the contribution is reasonable. Focusing on the border of the safe set to reduce computational effort is a logical approach. Additive kernels seem to be able to enhance the performance of safe BO algorithms for high-dimensional systems. However, it remains unclear how to choose kernel hyperparameters in practice.

**Weaknesses:**

I see the following main weaknesses:

(W1) The embedding of related work is insufficient. There are lines of work, which seem closely related, but are not cited appropriately.

(W2) Theoretical results are partially imprecise. Some of the results have been known before and build on previous results, yet these are not cited.

(W3) In the empirical study, information is missing on how to apply the algorithm and how hyperparameters are chosen. The choice of heuristics is not sufficiently documented and discussed.

(W4) Safety violations in hardware experiments are not adequately discussed.

Below, I provide a more detailed discussion of these weaknesses, ordered by sections.


## Related work:

* In related work, Bottero et al. 2022 is missing. It would make sense to compare against this, as this work explicitly focuses on safe BO without expander sets.
* Furthermore, studies on safe/constrained BO for cascaded control systems, such as Khosravi et al. 2022, seem directly related but are not mentioned.


## Theoretical Results:

* Lemma 4.1 is a known result that can be found in e.g., Berlinet and Agnan-Thomas 2004 Theorem 5
* Theorem 4.2 is imprecise; further information on this can be found in Fiedler 2023 (Section 4)
* The theoretical results section clearly builds on Chowdhury and Gopalan 2017, Theorem 2, using their results for confidence bounds without referencing this paper.
* There are more recent results on confidence bounds in GP Regression, which lead to more conservative bounds and do not rely on the information gain. Information on this can be found in Whitehouse et al. 2024
* Theorem 5.1: The RKHS norm of the safety functions is bounded by $B$ , not the safety functions themselves

## Empirical Study

For the stage-wise approach it is unclear, when to go to the next stage. This can only be chosen through a heuristic. In the paper it remains unclear, how to choose this in a practical application.

In the synthetic experiments, only a noise free setting is evaluated. The results would be more relevant and would underline the contribution of the method more, if function evaluations would be noisy. In addition, this setting would also better represent real-world settings.

In the experimental evaluation, the choice of hyperparameters of the GP and the choice of $\beta$ is unclear. While in the theoretical derivations, the RKHS-norm bound and the information gain are used, in the experiments, $\beta = 2$ is used as a heuristic. This choice is common in other safe BO works; however, it is not made transparent in the paper and can only be found in the implementation. The use of this heuristic invalidates all proven safety guarantees. A detailed discussion on this can be found in Fiedler et al. 2024.

The choice and the long lengthscales in the kernel lead to safety violations. Typically, shorter lengthscales compared to the domain size are applied in safe BO applications. This limits performance while exploring but can lead to fewer safety violations.
It would be interesting to know when safety violations occurred in the hardware experiments. I assume this is mostly in the first exploration stage.

For the hardware experiments, it is unclear what $T_0$ is and if there is even a switch in the exploitation stage.

## Discussion

The safety violations that occur in the empirical results need to be more thoroughly discussed. In light of these, the claim in the conclusion that it "can be seamlessly integrated into real-world complex control applications." is too confident, having observed 39 safety violations in the hardware experiments.


## Minor comments:

Line 191 and 194: Definitions of safe set and expander sets in Sui 2015 and Sui 2018 are different
Line 206-207: This statement is wrong. Berkenkamp 2016 uses a Matern kernel; generally, in many SafeBO applications Matern kernels with $\nu = 3/2$ are used.
Line 333: Typo: Lipschitz continuous
Line 317-318: Grammar in Theorem 5.1 and Theorem 5.2 "with R-sub-Gaussian"


## References (not in the paper)

- Berlinet, A., & Thomas-Agnan, C. (2004). _Reproducing Kernel Hilbert Spaces in Probability and Statistics_. Springer Science & Business Media.
- Bottero, A., et al. (2022). Information-theoretic safe exploration with Gaussian processes. _Advances in Neural Information Processing Systems_, 35, 30707-30719.
- Chowdhury, S. R., & Gopalan, A. (2017). On kernelized multi-armed bandits. In _International Conference on Machine Learning_ (PMLR).
- Fiedler, C. (2023). Lipschitz and Hölder continuity in reproducing kernel Hilbert spaces. _arXiv preprint arXiv:2310.18078_.
- Fiedler, C., Menn, J., Kreisköther, L., & Trimpe, S. (2024). On safety in safe Bayesian optimization. _Transactions on Machine Learning Research_.
- Khosravi, M., et al. (2022). Safety-aware cascade controller tuning using constrained Bayesian optimization. _IEEE Transactions on Industrial Electronics_, 70(2), 2128-2138.
- Whitehouse, J., Ramdas, A., & Wu, S. Z. (2024). On the sublinear regret of GP-UCB. _Advances in Neural Information Processing Systems_, 36.

**Questions:**

Please address the mentioned weaknesses.

Additional questions on the hardware experiments:
- When does the exploration phase stop, when does the exploitation phase start?
- How were the GP hyperparameters chosen?
- Why and when do safety violations occur?
- How can $T_0$ be chosen in real-world applications?

---

> ### Author Response · Authors · 2024-11-20
>
> We thank the reviewer for the detailed review and affirmation of the paper's contribution, as well as the help in improving the paper's references. We address the weaknesses raised by the reviewer and answer the reviewer's questions below. The corresponding modifications in the paper have been highlighted in purple.
>
> ---
>
> ### **Comment 1**
>
> > “(W1) The embedding of related work is insufficient. There are lines of work, which seem closely related, but are not cited appropriately.”
>
> **Response:**
> We thank the reviewer for providing the articles, which we will discuss one by one and add to the updated PDF.
>
> 1. **The Information-Theoretic Safe Exploration (ISE) algorithm by Bottero et al. (2022):**
>    ISE replaces uncertainty-driven exploration with a more principled information-theoretic approach. By directly maximizing information gain, ISE achieves better data efficiency compared to SafeOpt, especially in settings with heteroskedastic noise where uncertainty measures alone are insufficient. ISE is designed for continuous domains and does not require discretization of the parameter space, which enhances its high-dimensional optimization capability compared to SafeOpt. However, in recent work from NeurIPS 2024, Hübotter et al. (2024) reported that ISE "leads to significantly worse performance on the simplest of problems".
>    We are currently reimplementing the algorithm and will try to test it on our real experiment.
>
> 2. **The article by Khosravi et al. (2023):**
>    This work aligns with the SafeOpt framework at the algorithmic level and serves as an application of SafeOpt in the industrial domain. However, it introduces several experimental innovations:
>    - **Barrier-like penalty term:** The cost function incorporates a safety metric that penalizes gains approaching unsafe thresholds by utilizing experimentally determined unsafe controller parameters.
>    - **Robust stopping criterion:** They propose a stopping criterion based on the constrained expected improvement (CEI). The algorithm terminates when the ratio between the current and maximum CEI across iterations remains below a predefined threshold for three consecutive iterations.
>    - **Critical gain detection:** Driven by experiments, this step employs Fourier analysis to preemptively identify unstable parameter ranges, ensuring system stability.
>
> ---
>
> ### **References**
>
> - Jonas Hübotter, Bhavya Sukhija, Lenart Treven, Yarden As, and Andreas Krause. Transductive Active Learning: Theory and Applications. [https://openreview.net/forum?id=tZtepJBtHg&referrer=%5Bthe%20profile%20of%20Jonas%20H%C3%BCbotter%5D(%2Fprofile%3Fid%3D~Jonas_H%C3%BCbotter1)](https://openreview.net/forum?id=tZtepJBtHg&referrer=%5Bthe%20profile%20of%20Jonas%20H%C3%BCbotter%5D(%2Fprofile%3Fid%3D~Jonas_H%C3%BCbotter1))

---

> ### Author Response · Authors · 2024-11-20
>
> ### **Comment 2**
>
> > “(W2) Theoretical results are partially imprecise. Some of the results have been known before and build on previous results, yet these are not cited.”
>
> **Response:**
> We will discuss the comments in this section one by one.
>
> 1. **“Lemma 4.1 is a known result that can be found in e.g., Berlinet and Agnan-Thomas 2004 Theorem 5.”**
>    We carefully compare our proof with Theorem 5 in Berlinet and Agnan-Thomas (2004) (referred to as “Theorem 5” below). On the surface, our proof is a special case of Theorem 5, where the kernels are specific (Gaussian) and there are $ d $ kernels instead of two. From a deeper analysis of the proof process:
>    - We explicitly and mathematically demonstrate that $ K $ is positive definite by showing that the sum of positive definite kernels remains positive definite. Theorem 5 relies on the property that the sum of reproducing kernels is itself positive definite.
>    - Regarding the construction of inner product and norm in $ \mathcal{H} $, in our case, since the functions $ f_i $ depend on different variables $ x_i $, the RKHSs $ \mathcal{H}_i $ are orthogonal, and the minimal norm simplifies to the sum of norms. Theorem 5 accommodates cases where $ \mathcal{H}_1 $ and $ \mathcal{H}_2 $ may not be orthogonal, hence the need for the minimal norm.
>    - In the proof of reproducing property, our proof leverages the explicit form of the Gaussian kernels and their reproducing properties, while Theorem 5 demonstrates that the reproducing property holds using the mappings $ v $ and $ v^{-1} $, and properties of $ \mathcal{N} $ and $ \mathcal{F} $.
>
>      In conclusion, our proof is specific to Gaussian kernels and orthogonal RKHSs, while Theorem 5 is general. However, our proof is more constructive and computational, while Theorem 5 uses abstract functional analysis tools. Although the conclusions are similar, the proofs are very different.
>
> 2. **“Theorem 4.2 is imprecise; further information on this can be found in Fiedler 2023 (Section 4).”**
>    We have carefully reviewed Section 4 in Fiedler (2023), but did not find any conclusions or proofs related to the Lipschitz-continuity of additive Gaussian kernels. We would greatly appreciate it if the reviewer could point out any inaccuracies in the proof of Theorem 4.2 of our paper, similar to the comments provided by reviewer R4HB.
>
> 3. **“The theoretical results section clearly builds on Chowdhury and Gopalan 2017, Theorem 2.”**
>    The reviewer refers to the premises of Theorems 5.1 and 5.2 from the paper by Chowdhury and Gopalan (2017). In fact, these premises are derived from Theorems 1 and 2 in Sui et al. (2018), which we have included in our citations. Theorems 1 and 2 in Sui et al. (2018) themselves reference the proof of Theorem 2 in Chowdhury and Gopalan (2017). We acknowledge that we only cited Sui et al. (2018) and not Chowdhury and Gopalan (2017). We thank the reviewer for pointing this out and helping us improve the citation of the paper.
>
> 4. **“There are more recent results […] can be found in Whitehouse et al. 2024.”**
>    We use the proof based on maximum information gain to facilitate a more intuitive comparison with previous safe Bayesian optimization algorithms, such as Sui et al. (2018). Nevertheless, we thank the reviewers for recommending new theoretical articles, which offer potential directions for advancing future research.
>
> 5. **“Theorem 5.1: The RKHS norm of the safety functions is bounded by $ B $, not the safety functions themselves.”**
>    We thank the reviewer for pointing out this typo. We have revised the description of Theorems 5.1 and 5.2 accordingly, combining the suggestions from reviewer R4HB.

---

> ### Author Response · Authors · 2024-11-20
>
> ### **Comment 3**
>
> > “(W3) In the empirical study, information is missing on how to apply the algorithm and how hyperparameters are chosen. The choice of heuristics is not sufficiently documented and discussed.”
>
> **Response:**
> As demonstrated in the implementation details, similar to SafeOpt, users only need to define the performance function and constraint functions, set the (additive) kernels according to the problem dimension, and then directly call the algorithm for optimization. In contrast, many other safe Bayesian optimization algorithms involve a more intricate setup process before they can be applied to new datasets or complex systems.
>
> As for the selection of various hyperparameters, similar to other Bayesian optimization algorithms and deep learning methods, to the best of our knowledge, it remains an open problem. Although alternative approaches are discussed in Duvenaud et al. (2011) and Fiducioso et al. (2019), their application to practical tasks is often more complex.
>
> In line with Kirschner et al. (2019), we manually selected reasonable values for the hyperparameters of the methods. Similarly, we did not conduct an exhaustive hyperparameter search. We will provide a detailed discussion of the hyperparameters used in this paper and the rationale behind their selection.
>
> ---
>
> 1. **Choice of $ T_0 $**
>    $ T_0 $ represents the number of iterations in the safe exploration stage. According to Theorem 5.1, $ T_0 $ is theoretically finite. However, in the practical control optimization tasks discussed in the paper, excessive exploration is unrealistic. For instance, in our hardware experiment, 100 iterations required approximately 5 hours, with most of the time spent on motor performance testing. In this case, we selected $ T_0 = 15 $, and our controller achieved stable and good performance after about 45 iterations.
>
>    A similar analysis was conducted by Bardou et al. (2024) in their author response. Although their paper discussed asymptotic behaviours, they limited the total number of iterations to the range between 100 and 200.
>
> ---
>
> 2. **Choice of variance and lengthscales in base kernels, and the choice of $ \beta $.**
>    The selection of Gaussian Process (GP) hyperparameters in our paper was primarily aimed at aligning with the baseline experiments to enable a fair comparison. We compared six baseline algorithms in the paper, three of which were from Kirschner et al. (2019). To ensure consistency, we used the same hyperparameters where possible, including the choice of $ \beta = 2 $. Although we employed a more complex approach to construct $ \beta $ in Theorems 5.1 and 5.2 for comparison with Sui et al. (2018), in practical applications, using $ \beta = 2 $ does not compromise the safety guarantee. This is because, regardless of the choice of $ \beta $, we enforce $ \ln > h $ to ensure high-probability safety.
>
>    In the hardware experiments, certain field-oriented control domain knowledge informed the selection of the variance and lengthscale of the base kernels. Field-oriented control comprises three loops, with varying impacts on motor response:
>    - The $ P $-gain and $ I $-gain of the speed loop have the greatest impact on motor response.
>    - The $ P $-gain and $ I $-gain of the $ q $-axis current loop have a moderate impact.
>    - The $ P $-gain and $ I $-gain of the $ i $-axis current loop, as the reference current remains at zero, have the smallest impact and primarily affect signal safety.
>
>    Accordingly, we set the variances of $ k_1 $ and $ k_2 $ to be higher, allowing the GP to model larger variations in the performance function along these dimensions. Conversely, the variances of $ k_5 $ and $ k_6 $ were set lower, reflecting their lesser contribution to the overall model. Since the optimization results were already excellent under this configuration, we did not further fine-tune the lengthscales. However, if fine-tuning were necessary, we believe that relatively smaller lengthscales should be selected for $ k_1 $ and $ k_2 $, and larger lengthscales for $ k_5 $ and $ k_6 $. Additionally, functions such as `kernel.variance.set_prior()` or `kernel.lengthscale.set_prior()` could be used for real-time fine-tuning.

---

> ### Author Response · Authors · 2024-11-20
>
> ### **Comment 4**
>
> > “(W4) Safety violations in hardware experiments are not adequately discussed -- Why and when do safety violations occur?”
>
> **Response:**
> We thank the reviewer for this insightful comment. Similar questions were raised in the comments of reviewer zLs8, to which we provided detailed responses that can be referenced. In summary, the 39 constraint violations in 5 runs observed in the hardware experiments did not render the system unsafe or unstable. Rather, they represent the exploration of poorly performing controllers, such as those with large overshoot, slow transient response, or unacceptable steady-state error.
>
> These violations primarily occurred during the exploration phase. This can be reduced by adjusting the hyperparameters of the base kernels or modifying the performance function to make it smoother. However, such adjustments may require more iterations, which could increase computational costs in practical applications.
>
> Moreover, compared to the baseline safe Bayesian optimization algorithms included in the experiments, our method exhibited the lowest number of constraint violations under similar hyperparameter settings. As noted by Sui et al. (2018), safe BO aims to ensure safety with high probability rather than guaranteeing 100% safety. Therefore, all safe BO methods based on confidence intervals inherently have a probability of constraint violation, regardless of how sophisticated the hyperparameter design may be. For example, constraint violations are also reported in Khosravi et al. (2022).

---

> ### Author Response · Authors · 2024-11-20
>
> ### **Comment 5**
>
> > **Question on "can be seamlessly integrated into real-world complex control applications":**
>
> **Response:**
> We appreciate the reviewer highlighting this point. The statement "can be seamlessly integrated into real-world complex control applications" underscores the algorithm's user-friendliness. Users simply need to define the performance and safety functions according to the task and configure the base kernels and main hyperparameters. The algorithm can then be applied without complex re-implementation.
>
> This is evident from the implementation details provided. Whether applied to synthetic simulations or hardware experiments, the overall implementation process remains the same. The only differences lie in the definition of the performance and safety functions, as well as the selection of base kernels and the main function hyperparameters.
>
> ---
>
> ### **Comment 6**
>
> > **Suggestion on additional experiments:**
> > “In the synthetic experiments, only a noise free setting is evaluated. The results would be more relevant and would underline the contribution of the method more, if function evaluations would be noisy. In addition, this setting would also better represent real-world settings.”
>
> **Response:**
> We thank the reviewer for the suggestion regarding noisy synthetic experiments. While we believe that the noisy hardware experiments sufficiently represent real-world settings, and baseline synthetic experiments in Kirschner et al. (2019) and Bardou et al. (2024) are also noise free, we are willing to incorporate noisy settings into the synthetic benchmarks for better illustration.
>
> As these experiments are time-consuming (especially experiments with DuMBO), we will update the results in another updated PDF once they are completed.
>
> ---
>
> We greatly appreciate the reviewer's insightful comments and the time the reviewer dedicated to providing detailed feedback. We trust that our revisions and responses clearly convey the intent of our work and address the concerns. If any issues or questions remain, please feel free to bring them up. We would be happy to provide any additional clarifications.

---

> > ### Comment · Reviewer_QXk1 · 2024-11-26
> > **Reply to authors' response**
> >
> > ### Comment 1:
> >
> > I appreciate that the authors included related work by Bottero et al., Khoshravi et al., and Chowdhury et al..
> >
> > However, the inclusion of related work and the discussion of existing results is still insufficient.
> >
> > For example, the addition to the pdf in line 063-064 is not really fitting. Fiedler et al 2023 only concerns the relation of RKHS and Hölder Continuity and Whitehouse et al provides insights on regret bounds in GP-UCB. In principle, this could be interpreted as "theoretical analysis", but these papers are not directly related to SafeOpt and should not be referenced in this part of the related work section.
> >
> > Furthermore, this becomes obvious in the comments:
> >
> > "in practical  applications, using $\beta = 2$ does not compromise the safety guarantee", the contrary is shown in Fiedler et al. 2024
> >
> > In Comment 6, the authors claim that they use the same noise-free setting as Kirschner et al. (2019). However, in this paper, the synthetic experimental setting includes noise:
> > "On all experiments we add Gaussian noise with standard deviation 0.2 to obtain a similar signal-noise ratio as on our real-world application."
> >
> > ### Comment 2:
> >
> > 1. This is still not a contribution
> > 2. In this Theorem, only pointwise Lipschitz continuity and not continuity of the whole map is shown. In Line 912 only Lipschitz continuity in the first argument is shown, while this implies in some sense Lipschitz continuity of the kernel, it is not precise refer to Fiedler 2024 Lemma 4.1 and Prop. 4.4.
> > 3. In Whitehouse et al. 2024 the main point is that tighter confidence bounds are provided that lead to less conservative SafeOpt-type algorithms, which could also be used in practice (with some assumptions) instead of using $\beta = 2$
> >
> > ### Comment 3:
> >
> > I appreciate the additional discussion on hyperparameter choices. A critical aspect in SafeBO algorithms, is in fact the choice of kernel hyperparameters (Fiedler et al. 2024) as overestimation of lengthscales leads to safety violations. The kernel hyperparameters are therefore critical hyperparameters for safety which are not trivial to choose in practice.
> > With T0, an additional hyperparameter compared to Bottero et al. 2022, Berkenkamp et al. 2023  is introduced.
> >
> > Comment of the authors:
> > *Although we employed a more complex*
> > *approach to construct in Theorems 5.1 and 5.2 for comparison with Sui et al. (2018) in practical applications using $\beta = 2$ does not compromise the safety guarantee.*
> >
> > The choice of $\beta = 2$ is a heuristic that invalidates all safety guarantees; while it can work in practice as shown in previous works, it rather encourages cautious exploration than providing any guarantee. Even in a well-specified setting with a correct kernel choice, $\beta =2$ can lead to many safety violations see Fiedler et al. 2024.
> >
> > As other works also use this setting in their experiments, this choice can be made, but it is still a critical design choice, which should be transparent in the paper and not only be found in the code.
> >
> > ### Comment 4:
> >
> > We appreciate the discussion and this is in fact an interesting insight. However, I agree with Reviewer hSAs that the system might be not a good example for safe Bayesian optimization methods.
> >
> > ### Comment 5:
> >
> > The added discussion in the conclusion illustrates the issue nicely.
> >
> > ### Comment 6:
> >
> > See Comment 1; the setting in Kirschner et al. 2019 is not noise-free.
> >
> >
> >
> > While the changes in the paper are appreciated, I will not change my score.

---

> ### Author Response · Authors · 2024-11-27
> **Further explanation with additional experimental results**
>
> We are very grateful for the reviewer’s response.
>
> ---
> ### **Noisy Benchmark Environments**
>
> First, we would like to address the issue of the noisy environment. Although the experiment is not yet fully completed, we have included the existing results in the updated PDF; please refer to Appendix F. Since the optimization results of DuMBO and SafeCtrlBO are already very close to the maximum value of the function (under noise-free conditions), even Gaussian noise with a standard deviation of 0.0001 leads to negative regret in the Camelback function. In the Hartmann and Gaussian environments, the experimental results are very similar to those in the noise-free cases.
>
> Regarding the noisy setting in Kirschner et al. (2019), although the paper claims that Gaussian noise with a standard deviation of 0.2 is added, the official implementation uses the parameter setting ``noise_obs_mode: none''. Under this setting, the experimental results we obtained are consistent with those reported by Kirschner et al. (2019). However, when the standard deviation is set to 0.2 (indeed a large value), our reproduction results indicate that simple regret becomes negative, which differs from the results in Kirschner et al. (2019). In addition, the results of SwarmSafeOpt reported by Kirschner et al. (2019) are significantly lower than our reproduced results (in our reproduction, SwarmSafeOpt is comparable with LineBO). This suggests that Kirschner et al. (2019) may have used undeclared parameter choices.
>
> ---
> ### **Motor Cascade Control Example**
> Regarding the motor used in the experiment, we had further discussions with reviewer hSAs, and then reviewer hSAs raised the score. Reviewer hSAs initially thought that the motor would not encounter unsafe situations, but we explained that such cases did occur during the experiment. Currently, reviewer hSAs suggests that the FOC method for the motor does not require iterative adjustment. We provided reasonable explanations for this point as well, which are reflected in our latest conversation.
>
> ---
> ### **Hyperparameters**
>
> We are very grateful for the reviewer’s explanation regarding the choice of hyperparameters. However, keeping the hyperparameters consistent with the baseline methods (Kirschner et al. (2019) and Bardou et al. (2024)) ensures a fair comparison. For the additional hyperparameter $ T_0 $, we have also explained the rationale behind its selection. We have added these contents to the updated PDF, please refer to Appendix B.

---

### Official Review · Reviewer_R4HB · 2024-11-02

**Soundness:** 2
**Presentation:** 3
**Contribution:** 2
**Rating:** 3
**Confidence:** 5

**Summary:**

The authors propse a novel algorithm for safe Bayesian optimization that exploits purported properties of the squared-exponential kernel to facilitate the expansion of the safe set. The proposed idea is interesting and the authors report good theoretical results. However, the theoretical exposition is poor and the corresponding proofs are either incomplete or incorrect. Though I recommend rejection for this reason, I am open to changing my score if the authors improve the paper accordingly.

**Strengths:**

The paper is well-motivated and written in a very clear form. It also provides an adequate review of related works. The proposed algorithm is an interesting contribution and shows good empirical results.

**Weaknesses:**

The theoretical results are mostly stated usinng plain text instead of mathematical formulas, which makes them somewhat ambiguous and hard to understand at times.

The proofs of the theoretical results are incomplete and potentially wrong.

**Questions:**

The statement "Srinivas et al. (2010) demonstrated that BO methods can converge to the global optimum of unknown performance functions in fewer steps compared to genetic algorithm." is incorrect. Nothing in the paper compares their approach to a genetic algorithm, and there is no discussion on this.

Similarly, the statement "However, SAFEOPT uses Gaussian kernels as the covariance function of the Gaussian processes, which is effective mainly for low-dimensional problems (Bengio et al., 2005)" is inaccurate. Berrkenkamp et al. (2016) use a Matern kernel and there is nothing in Sui et al. (2015) that indicates that a squared-exponential kernel is necessary.

Is equation (1) correct? Shouldn't the last term have t in the subscript instead of k? The same holds for equation 2.

I feel that theorems 4.1 and 4.2 are mostly trivial and should be placed in the appendix.

Does line 5 in the pseudocode simply mean that the algorithm picks the point with maximal variance in B_n? If so, why not just write sigma insteadl of un-ln?

The presentation of outermost evaluated safe points is somewhat confusing. Is a_{oes} in the dataset? Would it help to introduce the data set \mathcal{D}_n and to write a_{oes} \in \mathcal{D}_n?

Theorems 5.1 and 5.2 are unclear to me. What do the authors mean by the "maximum allowable uncertainty for the exploration to converge to an ( \epsilon )-reachable safe region." and "Maximum allowable uncertainty for the performance function to converge to a ( \zeta )-optimal function value."?

In the proof of Lemma C.1, the step from line 878 to 880 is incomplete. To show that the posterior variance is indeed increasing, the authors also need to show that [K_t^{-1} k_t(x)]_i is positive, which the authors do not do.

I might have missed something, but I am not sure if the statement in line 894 holds. Take, for example, the points x_sb =1, x_oes = 0 and x_i = 10. For \lambda=0, we have || x(\lambda) - x_i ||_2 = 10 > || x_sb - x_oes||_2 = 1, yet the inner product (x(\lambda) - x_i ) (x_sb-x_oes)  = (0-10 ) (1-0) <0 is negative.

---

> ### Author Response · Authors · 2024-11-20
>
> We sincerely thank the reviewer for the time and effort dedicated to evaluating our manuscript, especially for providing an in-depth understanding of the theoretical results. Your perceptive comments and valuable suggestions have significantly improved the quality and clarity of our paper. Below, we address the concerns raised and respond to the reviewer’s questions. The corresponding modifications in the paper have been highlighted in blue.
>
> ---
>
> ### **Comment 1**
>
> > “The theoretical results are mostly stated using plain text instead of mathematical formulas.”
>
> **Response:**
> We thank the reviewer for the suggestion and have made the presentation of the theoretical results as formal as possible. Please refer to the updated PDF.
>
> ---
>
> ### **Comment 2**
>
> > “The statement "Srinivas et al. (2010) […]" is incorrect.”
> > “Similarly, the statement "However, SAFEOPT […]" is inaccurate.”
>
> **Response:**
> We sincerely thank the reviewer for the valuable corrections to the related work section, and we have made the necessary revisions to the corresponding content in the paper. [1] derived the regret bounds for Gaussian Process (GP) optimization and established the relationship between the cumulative regret of GP-UCB and the maximum information gain. Since the exploitation stage of our method employs GP-UCB as the acquisition function, the proof of Theorem 5.2 directly builds upon the conclusions of [1].
> [2, 3] did not specify a particular type of kernel, instead referring broadly to "some kernels." [4] utilized the Matérn kernel with $ \nu = 3/2 $ in their experimental section and included the Gaussian kernel as an example in their official implementation. Similarly, [5] employed the Matérn kernel with $ \nu = 5/2 $. As $ \nu $ increases, the Matérn kernel approaches the Gaussian kernel, whereas at smaller $ \nu $ values, the Matérn kernel exhibits stronger locality.
>
> ---
>
> ### **Comment 3**
>
> > “Is equation (1) correct? Shouldn't the last term have t in the subscript instead of k? The same holds for equation 2.”
>
> **Response:**
> In the context of control, signals, and systems, $ k $ is conventionally used to denote the time step in discrete systems or signals, while $ t $ is used for the time variable in continuous systems or signals. Discrete controllers are typically implemented using digital processors, microprocessors, or computers, whereas continuous controllers are generally realized using analog electronic components, such as capacitors, resistors, and operational amplifiers.
> Since the hardware experiments in this paper, as well as in related works (e.g., drone control [4] or robot control), involve discrete controllers, equations (1) and (2) adopt $ k $ to represent the time step. This choice aligns with the practical implementation of discrete control systems in digital hardware.
>
> ---
>
> ### **Comment 4**
>
> > “I feel that Theorems 4.1 and 4.2 are mostly trivial and should be placed in the appendix.”
>
> **Response:**
> We thank the reviewer’s suggestion and have moved Theorems 4.1 and 4.2 to the Appendix.
>
> ---
>
> ### **References**
>
> 1. Niranjan Srinivas, Andreas Krause, Sham M. Kakade, and Matthias Seeger. Gaussian process optimization in the bandit setting: no regret and experimental design. In *Proc. of the International Conference on Machine Learning (ICML)*, pp. 1015–1022, 2010.
> 2. Yanan Sui, Alkis Gotovos, Joel Burdick, and Andreas Krause. Safe exploration for optimization with gaussian processes. In *Proc. of the 32nd International Conference on Machine Learning (ICML)*, pp. 997–1005, Lille, France, 2015.
> 3. Yanan Sui, Vincent Zhuang, Joel Burdick, and Yisong Yue. Stagewise safe bayesian optimization with gaussian processes. In *Proc. of the 35th International Conference on Machine Learning (ICML)*, pp. 4781–4789, Stockholm, Sweden, 2018.
> 4. Felix Berkenkamp, Angela P. Schoellig, and Andreas Krause. Safe controller optimization for quadrotors with gaussian processes. In *Proc. of 2016 IEEE International Conference on Robotics and Automation (ICRA)*, pp. 491–496, Stockholm, Sweden, 2016.
> 5. Marcello Fiducioso, Sebastian Curi, Benedikt Schumacher, Markus Gwerder, and Andreas Krause. Safe contextual bayesian optimization for sustainable room temperature pid control tuning. In *Proc. of the Twenty-Eighth International Joint Conference on Artificial Intelligence (IJCAI-19)*, pp. 5850–5856, 2019.

---

> ### Author Response · Authors · 2024-11-20
>
> ### **Comment 5**
>
> > “Does line 5 in the pseudocode simply mean that the algorithm picks the point with maximal variance in $ B_n $? If so, why not just write $ \sigma $ instead of $ u_n - l_n $? ”
>
> **Response:**
> Yes, line 5 of the pseudocode represents the acquisition function used during the safe exploration stage, which selects the point in $ B_n $ where the performance function exhibits the greatest uncertainty. While this could be expressed directly in terms of $ \sigma $, we chose to write it in the form of the upper confidence bound minus the lower confidence bound to maintain consistency with line 11 and to emphasize the distinction between the acquisition functions used in the exploration and exploitation phases.
>
> ---
>
> ### **Comment 6**
>
> > “Is $ a_{\text{oes}} $ in the dataset? Would it help to introduce the dataset $ \mathcal{D}_n $ and to write $ a_{\text{oes}} \in \mathcal{D}_n $? ”
>
> **Response:**
> If the "dataset" referred to by the reviewer corresponds to the set of observations $ D_n $, then $ a_{\text{oes}} \in D_n $. According to line 257, $ a_{\text{oes}} \in S_n^{\text{eval}} $, where $ S_n^{\text{eval}} \subseteq S_n $ and $ S_n^{\text{eval}} $ contains all the evaluated points in $ S_n $. If all points in $ D_n $ are safe observations, then $ D_n = S_n^{\text{eval}} \subseteq S_n $.
> $ a_{\text{oes}} $ represents the point(s) in $ S_n^{\text{eval}} $ that is closest to a specific $ a_{\text{sb}} $. For example, in a 1D case, suppose the current $ S_n^{\text{eval}} = {-1, 2, 3} $ and $ a_{\text{sb}} = -3 $ and $ 5 $. Then, $ a_{\text{oes}} = -1 $ and $ 3 $, and the outermost region is $ [-3, -1] \cup [3, 5] $.
>
> ---
>
> ### **Comment 7**
>
> > “What do the authors mean by the "maximum allowable uncertainty for the exploration to converge to an $ \epsilon $-reachable safe region." and "Maximum allowable uncertainty for the performance function to converge to a $ \zeta $-optimal function value."?”
>
> **Response:**
> The maximum allowable uncertainty, $ \epsilon $, for the exploration to converge to an $ \epsilon $-reachable safe region is a very small positive number used to define the stopping criterion for the exploration stage. During the early stages of the safe exploration, the safe set $ S_t $ is small, the outermost region $ O_t $ is large, and the uncertainty $ \sigma_t $ in the performance function and safety function values $ J $ and $ G_i $ at most safe boundary points $ a_{\text{sb}} $ in $ B_t $ is high.
>
> As the safe exploration progresses, the safe set $ S_t $ gradually expands, and the uncertainty in $ J $ and $ G_i $ for each $ a_{\text{sb}} $ decreases. By a certain iteration $ t^* $, the uncertainty in $ J $ and $ G_i $ for all $ a_{\text{sb}} $ becomes less than or equal to a predefined small positive number $ \epsilon $. This indicates that the safe region (the region of the performance function and safety function values above the safe threshold) has been sufficiently explored, marking the end of the safe exploration phase.
>
> Similarly, the maximum allowable uncertainty, $ \zeta $, for the performance function to converge to a $ \zeta $-optimal function value is a very small positive number used to define the signal that the exploitation stage has converged to the optimal performance function value. At the beginning of the exploitation stage, the difference between the optimized best function value $ f(a_{\text{opt}}) $ and the theoretical optimal value $ f(a^*) $ is large.
>
> As exploitation progresses, $ f(a_{\text{opt}}) $ gradually approaches $ f(a^*) $, so the simple regret $ r_t = f(a^*) - f(a_{\text{opt}}) $ decreases. After a certain iteration $ t^* $, $ r_t $ becomes less than or equal to a predefined small positive number $ \zeta $, indicating that the difference between $ f(a_{\text{opt}}) $ and $ f(a^*) $ is sufficiently small. At this point, it can be considered that the optimal value of the function has been effectively obtained.
>
> ---

---

> ### Author Response · Authors · 2024-11-20
>
> ### **Comment 8**
>
> > “In the proof of Lemma C.1, the step from line 878 to 880 is incomplete.”
>
> **Response:**
> We appreciate this insightful question very much, as it helps us to correct one mistake in the proof of Lemma C.1 (Lemma D.2 in the updated PDF). We have modified and improved the proof and please refer to the updated PDF for the revision.
>
> The confusion in the original proof arose from a typographical error, where we mistakenly wrote $ \mathbf{k}_t(x) $ as $ [\mathbf{k}_t(x)]_i $, inadvertently transforming a vector into a scalar. After correcting this mistake, we observe that $ \mathbf{K}_t $ is a symmetric positive definite (PD) matrix, and consequently, $ \mathbf{K}_t^{-1} $ is also PD. This allows us to conclude that $ \mathbf{k}_t(x)^\top \mathbf{K}_t^{-1} \mathbf{k}_t(x) > 0 $.
>
> Furthermore, since $ -2 \times -\frac{d_i}{l^2} \geq 0 $, we can deduce the result in line 880:
> $$
> \frac{\partial \sigma_t^2(x)}{\partial d_i} \geq 0.
> $$
>
> ---
>
> ### **Comment 9**
>
> > “I am not sure if the statement in line 894 holds. Take, for example, the points $ x_{\text{sb}} = 1 $, $ x_{\text{oes}} = 0 $ and $ x_i = 10 $. For $ \lambda=0 $, we have $ || x(\lambda) - x_i ||_2 = 10 > || x_{\text{sb}} - x_{\text{oes}}||_2 = 1 $, yet the inner product $ (x(\lambda) - x_i ) (x_{\text{sb}}-x_{\text{oes}}) = (0-10 ) (1-0) <0 $ is negative.”
>
> **Response:**
> In the case provided by the reviewer, where $ x_{\text{sb}} = 1 $ and $ x_{\text{oes}} = 0 $, any evaluated safe point $ x_i $ must lie within $ [-\infty, 0] $. This is because $ x_{\text{oes}} = 0 $ is already the closest evaluated safe point to $ x_{\text{sb}} $. In this scenario, the outermost region is $ [0, 1] $, while $ [1, \infty] $ represents the current unsafe region. Therefore, $ x_i = 10 $ is not possible under these conditions.
>
> As stated in line 894 of the paper, $ x_{\text{sb}} $ is farther from $ x_i $ than $ x_{\text{oes}} $ is. The positional relationship of these points in the 1D case can be described as:
>
> $\text{unsafe region}$ $\cdots$ $x_{\text{sb,left}}$ $\cdots$ $x(\lambda_{\text{left}})$ $\cdots$ $x_{\text{oes,left}}$ $\cdots$ $x_i$ $\cdots$ $x_{\text{oes,right}}$ $\cdots$ $x(\lambda_{\text{right}})$ $\cdots$ $x_{\text{sb,right}}$ $\cdots$ $\text{unsafe region}$.
>
> any evaluated safe point $ x_i $ must lie between the two outermost evaluated safe points $x_{\text{oes,left}}$ and $x_{\text{oes,right}}$. This ensures that the sign of $ x(\lambda) - x_i $ is identical to that of $ x_{\text{sb}} - x_{\text{oes}} $ on the same side (either left or right, depends on $ x(\lambda) $), guaranteeing the following inequality holds:
>
> $$
> \left( x(\lambda) - x_i \right)^\top \left( x_{\text{sb}} - x_{\text{oes}} \right) \geq 0.
> $$
>
> ---
>
> If any issues or questions remain, please feel free to bring them up. We would be happy to provide further clarifications.

---

> > ### Comment · Reviewer_R4HB · 2024-11-20
> >
> > Thank you for responding to my questions. The updated Lemma (Lemma D.2) is seemingly incorrect. The authors state ∂kt(x)/∂di
> >  =−d_i/ l^2* kt(x), which is incorrect, unless I misunderstood the definition of k_t(d_i). In fact, ∂kt(x)/∂di = −d_i/ l^2* (0, 0, 0, ... k(x, x_i) , ..., 0 ,0 ). Though this potentially does not affect, the final result, I suggest the authors carefully introduce the notation employed here and revise the result accordingly to avoid confusion.

---

> > > ### Author Response · Authors · 2024-11-27
> > > **A gentle reminder for new explanations and updated PDF**
> > >
> > > Dear Reviewer,
> > >
> > > This is a gentle reminder that we have added new content to address your concerns in the updated PDF. Please refer to Theorem D.3 and Appendix E. Thank you again for helping us revisit the functionality of our proposed acquisition function.

---

> ### Author Response · Authors · 2024-11-20
>
> We sincerely thank the reviewer for the very careful review. To reduce confusion, we have rewritten the proof as Lemma D.2 and Theorem D.3. Please refer to the updated PDF. We added a premise that Theorem D.3 holds when $ O_n $ is sufficiently large, corresponding to the early safe exploration stage. In the later stage of safe exploration, or when the problem dimension increases, $ O_n $ becomes smaller accordingly, and the result of Theorem D.3 may become inaccurate. This may explain the results of the ablation study in Appendix E, where SafeCtrlBO gradually outperforms StageOpt in 6D and 10D as the number of iterations increases when the same kernel is used.
>
> The key difference between SafeCtrlBO and StageOpt in implementation is that SafeCtrlBO uses the additive Gaussian kernel and employs a new acquisition function. If, as we have proven earlier, the acquisition results of the new acquisition function are equivalent to those in StageOpt, then the results of SafeCtrlBO in the ablation study should be nearly the same as those of StageOpt (because here the additive Gaussian kernel is not used). This was initially a source of confusion for us as well. We thank the reviewer for raising this question, which prompts us to revisit the functionality of our proposed acquisition function. We hope that this clarification highlights the contributions of our paper more effectively.

---

### Official Review · Reviewer_hSAs · 2024-11-03

**Soundness:** 3
**Presentation:** 3
**Contribution:** 3
**Rating:** 6
**Confidence:** 2

**Summary:**

In this paper, the authors propose a safe Bayesian optimization framework utilizing Gaussian processes with additive squared exponential functions. The motivation for the new kernel function is the poor performance of the "standard" SE kernel for high-dimensional spaces. The authors argue that using BO for controller tuning often requires operating in high-dimensional spaces due to a large number of parameters of the controllers to be considered. It is experimentally evaluated that the proposed algorithm, SafeCtrlBo, can lead to better results than other safe BO algorithms. Furthermore, the authors present theoretical results on the finite-time convergence of the algorithm.

The main contribution is utilizing additive SE-kernels in a Bayesian optimization framework and proofing the finite-time convergence.

UPDATE: score raised

**Strengths:**

- The paper addresses an important problem of (safe) BO, which is the limitation to low-dimensional problems.
- Using additive kernel functions in a safe BO setting to increase its performance in high dimensional problems seems, to the best of my knowledge, a novel idea. The results indicate that this approach can be superior to "standard" safe BO approaches for some systems.
- To the best of my knowledge, the math seems to be sound and the proofs are correct.

**Weaknesses:**

1. The main motivation for introducing the new kernel function is its superior in high dimensions. However, all evaluations in the paper are still pretty low-dimensional, with dimensions up to 10. In fact, in the related work it is mentioned that existing safe BO approaches work with systems where "three controller [...] each controller having only two parameters". However, the proposed algorithm is experimentally evaluated on the exact same numbers of parameters.

2. I agree that controller tuning including many parameters can be tricky. However, cascaded controllers are a bad example of that. In practice, these structures can be tuned very efficiently as you start with the inner loop until a specific performance is reached, then the next loop, and so on. All parameters are very meaningful, and the process is quite transparent. Furthermore,
motor controllers are a bad example for safe BO. As long as no load is attached to the motor (something you don't do for tuning the controllers), it is almost impossible to destroy the system. Typical amplifiers do have a "max current" setting, or it is simply defined in the software. Therefore, I cannot understand the safety concerns that the authors mentioned for the experiment.

Long story short: Although the adapted safe BO method sounds interesting, the motivation with cascade controllers and the experimental evaluation are not a good choices.

**Questions:**

1: Can you include an experiment with a truly high-dimensional system?

2: As mentioned in the weakness section, I think cascade controllers are not a good example. I suggest something like distributed controllers in large water/power networks. Could you test the algorithms on more complex systems?

---

> ### Author Response · Authors · 2024-11-20
>
> We sincerely thank the reviewer for the time and effort dedicated to evaluating our manuscript. Your perceptive comments and valuable suggestions have greatly improved the quality and clarity of our paper. Below, we address the concerns raised and respond to the reviewer’s questions. The corresponding modifications in the paper have been highlighted in brown.
>
> ---
>
> ### **Comment 1**
>
> > “The main motivation for introducing the new kernel function is its superior in high dimensions. However, all evaluations in the paper are still pretty low-dimensional, with dimensions up to 10.”
> > “Can you include an experiment with a truly high-dimensional system?”
>
> **Response:**
> As outlined in lines 083-084 of the paper, the control problems studied here lie within the low-to-moderate dimensional range, as opposed to the high-dimensional settings considered in [1] or [2]. The experimental results demonstrate that, at the problem scale studied in this work, traditional safe Bayesian optimization algorithms are prone to being trapped in local optima, whereas high-dimensional safe Bayesian optimization methods, such as LineBO, require more iterations to achieve competitive performance compared to the proposed method.
>
> The limitations of our method for "truly high-dimensional problems" are also discussed in this paper. As described in lines 536-539 of the paper (currently in lines 508-516 in the updated PDF), the number of orders of additive Gaussian kernels equals the dimension of the problem. For instance, a 100-dimensional problem would involve 100 orders of additive kernels. Designing and combining these kernel components effectively requires extensive domain knowledge and substantial experimental effort. Furthermore, the computational cost associated with these combinations becomes a significant barrier to scaling this approach to very high-dimensional problems. Similar insights are drawn in Sections 3.4 and 4 of [3].
>
> ---
>
> ### **Comment 2**
>
> > “In fact, in the related work it is mentioned that existing safe BO approaches work with systems where "three controller [...] each controller having only two parameters". However, the proposed algorithm is experimentally evaluated on the exact same numbers of parameters.”
>
> **Response:**
> The description of the related work [4] in this question is in lines 066-067 of the paper. In the related work, the quadrotor is equipped with three motion controllers, each responsible for controlling movement along one of the \(x\), \(y\), and \(z\) axes. Each axis has two control parameters: proportional gain and derivative gain. Importantly, the performance functions associated with the three axes are independent, meaning adjustments to the \(x\)-axis controller do not influence the performance of the \(y\) or \(z\) axes.
> In short:
> - Three performance functions, each influenced by 2 parameters, so **only 2 parameters are tuned simultaneously**.
>
> In contrast, although the hardware experiments conducted in this paper also involve three controllers, each with two parameters, a key distinction is that the six control parameters collectively determine a **single performance function**. Consequently, all six parameters need to be optimized simultaneously to make this unique performance function optimal.
> In short:
> - Only one performance function, influenced by 6 parameters, so **6 parameters are tuned simultaneously**.
>
> ---
>
> ### **References**
>
> 1. Johannes Kirschner, Mojmir Mutny, Nicole Hiller, Rasmus Ischebeck, and Andreas Krause. Adaptive and safe Bayesian optimization in high dimensions via one-dimensional subspaces. In *Proc. of the 36th International Conference on Machine Learning (ICML)*, pp. 3429–3438, 2019.
> 2. Anthony Bardou, Patrick Thiran, and Thomas Begin. Relaxing the additivity constraints in decentralized no-regret high-dimensional Bayesian optimization. In *ICLR ’24: International Conference on Learning Representations (ICLR)*, 2024.
> 3. David K Duvenaud, Hannes Nickisch, and Carl Rasmussen. Additive Gaussian processes. In *Advances in Neural Information Processing Systems*, volume 24, 2011.
> 4. Felix Berkenkamp, Angela P. Schoellig, and Andreas Krause. Safe controller optimization for quadrotors with Gaussian processes. In *Proc. of 2016 IEEE International Conference on Robotics and Automation (ICRA)*, pp. 491–496, Stockholm, Sweden, 2016.

---

> ### Author Response · Authors · 2024-11-20
>
> ### **Comment 3**
>
> > “I agree that controller tuning including many parameters can be tricky. However, cascaded controllers are a bad example of that.”
> > “I suggest something like distributed controllers in large water/power networks.”
>
> **Response:**
> The reviewer's comment regarding cascade controller tuning is accurate in the context of traditional industrial applications. However, achieving precise control with this approach typically requires iterative adjustments, progressing sequentially from the innermost to the outermost layer. Since parameter changes in the outer controllers can affect the optimality of the inner controllers, it becomes necessary to cycle back to the innermost layer for re-tuning, followed by further adjustments towards the outermost layer. This iterative process continues until the system's performance reaches a near-optimal state. To enhance optimization efficiency, simultaneous tuning of both inner and outer controllers becomes essential, allowing for precise control of cascade structure controllers.
>
> Furthermore, it is worth noting that in the inner loop, there are two controllers operating in parallel rather than in a cascade configuration. This distinction makes our problem more challenging compared to typical cascade control.
>
> In addition, as noted in lines 533-535 of the paper (currently in lines 505-507 in the updated PDF), the proposed algorithm is not constrained to cascade controllers. We highlighted various other examples of multi-controller systems in lines 29-34 and lines 84-89. The distributed controller systems mentioned by the reviewer are also excellent examples. Due to equipment limitations, our experiments were conducted on a permanent magnet synchronous motor (PMSM) platform. Nevertheless, we are optimistic that future opportunities will allow us to apply the algorithm to larger and more complex control platforms, as suggested by the reviewer.
>
> ---
>
> ### **Comment 4**
>
> > “Furthermore, motor controllers are a bad example for safe BO. As long as no load is attached to the motor […] it is almost impossible to destroy the system. Typical amplifiers do have a "max current" setting, or it is simply defined in the software.”
>
> **Response:**
> In fact, although the permanent magnet synchronous motor (PMSM) operates in a no-load state during controller parameter tuning, certain controller parameters will still make the entire system unstable during our experiment. In such cases, the speed tracking error increases over time, requiring human intervention to forcibly shut down the system. Safe BO algorithms can prevent the selection of such parameters, thereby ensuring the system remains in a stable state throughout the process.
>
> While setting saturation limits for controller outputs, currents, or voltages in the software can also mitigate instability, this approach relies on specific domain knowledge. Moreover, implementing such saturation constraints effectively alters the distribution of the performance function. For instance, when the control parameters reach certain thresholds, the controller output saturates, and further increases in control parameter values no longer affect the performance function. This manual modification of the performance landscape can pose challenges for safe BO algorithms, potentially making it harder for them to converge to the optimal parameters.
>
> ---
>
> We sincerely appreciate your insightful comments and the time invested in providing detailed feedback. We hope our revisions and responses effectively convey the intent of our work and address your concerns. If any issues or questions remain, please do not hesitate to raise them—we are more than happy to offer further clarifications.

---

> > ### Comment · Reviewer_hSAs · 2024-11-27
> >
> > Thank you for the detailed response. Now I understand that the 6 parameters are tuned simultaneously whereas the 3x2 parameters for the quadcopters are optimized independently. I still believe that cascade control is not a good example for the approach. In motor control, the design of the cascade is typically not an iterative process, instead there are clear rules how to tune each stage. Not saying that this example is useless but there are much better systems, which would strengthen the paper.
> >
> > However, I’m satisfied by some of the authors updates and will raise my score.
> >
> > Thanks to the authors for their work and I’ll looking forward to the future work.

---

> > > ### Author Response · Authors · 2024-11-27
> > >
> > > We are very grateful for the reviewer's response!
> > >
> > > For most drive motors, as long as parameters such as stator resistance, d-q axis inductance $ L_d $, $ L_q $, rotor inertia, number of pole pairs, and others are known, the control parameters of the three PI controllers in FOC can be calculated using established formulas (derived from the approximate linear model). However, due to the limitations of linear approximations, the control parameters obtained in this manner typically result in nearly satisfactory motor performance but are not "optimal". In this experiment, we do not have precise values for these motor parameters, but our goal is to determine control parameters that optimize the current task's performance. Thus, the parameter tuning process is iterative.
> > >
> > > However, we sincerely appreciate the future research direction suggested by the reviewer. In future work, we plan to explore more complex and widely used robotic systems, such as gantry systems, and to conduct simulation experiments on large power systems.

---

### Author Response · Authors · 2024-12-03
**Summary of Official Comments by Authors**

We sincerely thank each reviewer for their valuable comments. We have revised the paper based on all the feedback provided.

First, we thank the reviewers for recognizing the ideas and experimental results presented in the paper. Reviewer R4HB expressed particular interest in the theoretical aspects. We provided detailed responses and revised the paper accordingly, especially updating Theorem D.3, which aligns well with the results of the ablation study in Appendix E. We believe this revision significantly enhances the quality of the paper.

Reviewer QXk1 expressed concerns about the selection of hyperparameters in the experiments. We addressed this by combining insights from relevant literature and the specific experimental setup, and we incorporated these explanations into the updated paper. Additionally, we included benchmark experiments in noisy environments as requested, further improving the experimental results.

Reviewers zLs8 and QXk1 expressed concerns about constraint violations in the hardware experiments. We provided detailed explanations and included these clarifications in the paper. Reviewer hSAs raised concerns regarding the dimensionality of the research problem and offered suggestions on the experimental equipment. We addressed these points in detail and incorporated the necessary revisions, and we are pleased to note that we have successfully clarified reviewer hSAs's questions.

We believe that these revisions have greatly improved the paper's readability and made the contributions of this work much clearer.

---

### Meta-Review · Area_Chair_Lgqd · 2024-12-23

**Metareview:**

This paper investigate a SafeControlBO method to optimize multiple controllers at the same time. It shows improvements addressing the challenges associated with tuning complex control systems that involve multiple parameters while keeping online safety.
Strengths of the paper: effectively optimization while ensuring safety. Its empirical validation demonstrating superior performance compared to some existing safe Bayesian optimization methods. The use of additive kernels enhances efficiency, making it suitable for complex control systems.
Weaknesses: incremental theoretical improvements, hyperparameter tuning issues.
The AC likes the paper in general, but the reviewers opinions make sense, this paper should be improved to reach the bar of ICLR.

**Additional Comments On Reviewer Discussion:**

After evaluated the quite disparate scores by reviewers, the AC realized that the opinions by the reviewers are not as different as shown by the score. It is an interesting paper while still need substantial improvement before reaching the bar of ICLR 2025.

---

### Decision · Program_Chairs · 2025-01-22

Reject